# SPARSE COVARIANCE NEURAL NETWORKS

## ABSTRACT

Covariance Neural Networks (VNNs) perform graph convolutions on the covariance matrix of tabular data and achieve success in a variety of applications. However, the empirical covariance matrix on which the VNNs operate may contain many spurious correlations, making VNNs' performance inconsistent due to these noisy estimates and decreasing their computational efficiency. To tackle this issue, we put forth Sparse coVariance Neural Networks (S-VNNs), a framework that applies sparsification techniques on the sample covariance matrix before convolution. When the true covariance matrix is sparse, we propose hard and soft thresholding to improve covariance estimation and reduce computational cost. Instead, when the true covariance is dense, we propose stochastic sparsification where data correlations are dropped in probability according to principled strategies. We show that S-VNNs are more stable than nominal VNNs as well as sparse principal component analysis. By analyzing the impact of sparsification on their behavior, we provide novel connections between S-VNN stability and data distribution. We support our theoretical findings with experimental results on various application scenarios, ranging from brain data to human action recognition, and show an improved task performance, stability, and computational efficiency of S-VNNs compared with nominal VNNs.

## 1 INTRODUCTION

Covariance-based data processing is key to machine learning pipelines due to its ability to whiten data distributions, identify principal directions, and estimate the interdependencies among features. Such advantages have been shown in several applications including brain connectivity estimation (Bessadok et al., 2022; Qiao et al., 2016), financial data (de Miranda Cardoso et al., 2020; Wang & Aste, 2022) and human action recognition through motion sensors measurements (Liao et al., 2022; Wang et al., 2023). One prominent approach is Principal Component Analysis (PCA), which maximizes the variance of data points by projecting them onto the eigenvectors of their covariance matrix (Jolliffe & Cadima, 2016). However, PCA-based approaches are unstable to errors in the covariance estimation, i.e., a poor estimate of the covariance matrix and its eigenvectors may lead to unpredictably bad results, especially in high-dimensional and low-data regimes or when the covariance matrix eigenvalues are close to each other, and these limitations also appear in more advanced PCA variants such as kernel PCA (Jolliffe, 2002; Paul, 2007; Baik et al., 2005; Schölkopf et al., 1997). To overcome this issue, coVariance Neural Networks (VNNs) have been proposed (Sihag et al., 2022). VNNs apply graph convolutions to the covariance matrix, an operation that, similarly to PCA, learns the importance of the principal directions of the data. Due to their graph convolution operation, they inherit the stability property of GNNs (Gama et al., 2020b) and extend it to errors in covariance estimation, which guarantees more reliable performance even in the presence of poor covariance estimates since their output difference when operating on the sample and true covariance is bounded (Sihag et al., 2022; 2024; 2023). Nevertheless, VNNs operate on the sample covariance matrix, which results in two limitations. First, VNNs are sensitive to settings in which covariance estimation is problematic, especially in high-dimensional and low-data regimes (Paul, 2007; Baik et al., 2005; Féral & Péché, 2007). Second, VNNs are computationally expensive and memory inefficient as the sample covariance matrix is typically dense no matter whether the true covariance matrix is dense or sparse due to sample estimation errors, which is especially problematic on high-dimensional datasets.

To overcome these limitations, we propose sparsification-based covariance regularizers for VNNs to improve the covariance matrix estimation. We perform stability analysis for sparse VNNs (S-VNNs)

to investigate the effects of covariance sparsification techniques on the performance of VNNs. *When true covariance matrices are sparse*, we propose hard and soft thresholding strategies and show that S-VNNs are more robust to both the finite-data estimation error and the covariance sparsification thresholding. *For generic scenarios where true covariance matrices are not necessarily sparse*, we propose a stochastic sparsification approach and prove the stability of S-VNNs in this setting. Our findings quantify a trade-off between stability and sparsification (i.e., efficiency). That is, dropping covariances according to their absolute values leads to higher stability but, in some cases, low sparsification, whereas dropping a fixed percentage of covariance values improves sparsification at the expense of stability. We corroborate our findings with experiments on synthetic and real datasets highlighting the benefits of sparsity on covariance-based neural processing both in terms of performance and computation. Our specific contributions are as follows.

**(C1) Sparse covariance neural networks.** We propose covariance sparsification techniques for VNNs to improve performance and reduce computation. When the true covariance is sparse, we perform hard and soft covariance thresholding. When the true covariance is dense, we propose a stochastic sparsification that allows controlling the sparsification level and the desired stability.

**(C2) Stability to covariance sparsification.** We characterize the effects of covariance sparsification on the VNN through stability analysis. We show that S-VNNs can not only reduce computation but also improve stability under appropriate settings, compared to the nominal VNN and the sparse PCA.

**(C3) Empirical validation.** We validate our theoretical findings with experiments on real and synthetic datasets, demonstrating the importance of sparsification in low-data settings and the improved efficiency of S-VNNs on brain data and human action recognition use cases.

## 2 PROBLEM FORMULATION

Consider $t$ samples $\{\mathbf{x}_i\}_{i=1}^t$ of a random vector $\mathbf{x} \in \mathbb{R}^N$ with mean $\boldsymbol{\mu} = \mathbb{E}[\mathbf{x}]$, covariance $\mathbf{C} = \mathbb{E}[(\mathbf{x}-\boldsymbol{\mu})(\mathbf{x}-\boldsymbol{\mu})^\top]$ and respective estimates $\hat{\boldsymbol{\mu}} = \sum_{i=1}^t \mathbf{x}_i/t$, $\hat{\mathbf{C}} = \sum_{i=1}^t (\mathbf{x}_i - \hat{\boldsymbol{\mu}})(\mathbf{x}_i - \hat{\boldsymbol{\mu}})^\top/t$. The estimated covariance $\hat{\mathbf{C}}$ captures the data interdependencies and its eigendecomposition $\hat{\mathbf{C}} = \hat{\mathbf{V}}\hat{\boldsymbol{\Lambda}}\hat{\mathbf{V}}^\top$ with eigenvectors $\hat{\mathbf{V}} = [\hat{\mathbf{v}}_0, \dots, \hat{\mathbf{v}}_{N-1}] \in \mathbb{R}^{N \times N}$ and eigenvalues $\hat{\boldsymbol{\Lambda}} = \mathrm{diag}(\hat{\lambda}_0, \dots, \hat{\lambda}_{N-1})$ is used in principal component analysis (PCA), which performs the projection $\tilde{\mathbf{x}} = \hat{\mathbf{V}}^\top \mathbf{x}$. However, PCA is unstable as the estimated eigenvectors $\hat{\mathbf{V}}$ may differ significantly from the true ones $\mathbf{V}$ (where $\mathbf{C} = \mathbf{V}\boldsymbol{\Lambda}\mathbf{V}^\top$) especially if the true eigenvalues are close to each other (Jolliffe, 2002).

To overcome this, Sihag et al. (2022) proposed covariance filters as an extension of PCA that is stable to finite-sample estimation uncertainties even with close eigenvalues. Each data point is seen as a node of a graph and the data sample $\mathbf{x}$ as a graph signal, where the covariance $\mathbf{C}$ represents the graph structure. Covariance filters build on the standard graph convolutional filters, which shift a signal $\mathbf{x}$ over a graph represented by the shift operator $\mathbf{S}$ (e.g., adjacency matrix or Laplacian) according to the equation $\sum_{k=0}^K h_k \mathbf{S}^k \mathbf{x}$, where $K$ is the order of the filter and $h_k$ are learnable parameters (Gama et al., 2020c; Ortega et al., 2018; Isufi et al., 2024). Several popular GNNs, such as GCN (Kipf & Welling, 2016), are special instances of the generic graph convolutional filter. Building on this, the covariance filter of order $K$ and the $l$-th VNN layer, which assembles a filter bank of $F_{l-1} \times F_l$ covariance filters and a non-linear activation function $\sigma(\cdot)$, are defined as, respectively,

$$\mathbf{H}_{fg}^l(\hat{\mathbf{C}}) = \sum_{k=0}^K h_{klfg}\hat{\mathbf{C}}^k \ \text{ and } \ \mathbf{u}_f^l = \sigma\left(\sum_{g=1}^{F_{l-1}} \mathbf{H}_{fg}^l(\hat{\mathbf{C}})\mathbf{u}_g^{l-1}\right) \ f = 1, \dots, F_l, \ l = 1, \dots, L \quad (1)$$

where $\{\mathbf{u}_f^l \in \mathbb{R}^N\}_{f=1}^{F_l}$ are the outputs of the $l$-th VNN layer, each produced by the $f$-th covariance filter bank, which contains $F_{l-1}$ covariance filters processing each of the signals at the previous layer $\{\mathbf{u}_g^{l-1} \in \mathbb{R}^N\}_{g=1}^{F_{l-1}}$ separately. At the first layer, we have $\{\mathbf{u}_g^0 = \mathbf{x}_g\}_{g=1}^{F_0}$ where $F_0$ is the node feature size. This operation is related to PCA as it processes the eigenvectors of the covariance matrix (see Appendix A for details). We denote the VNN architecture as $\Phi(\mathbf{x}, \hat{\mathbf{C}}, \mathcal{H})$, where $\mathcal{H} = \{h_{klfg}\}_{klfg}$ contains all network parameters for each order $k$, layer $l$, input signal $g$ and output signal $f$. The output of the last layer $\mathbf{u}^L = \Phi(\mathbf{x}, \hat{\mathbf{C}}, \mathcal{H})$ contains the final representations generated by VNN and can be directly used for a downstream task (e.g., classification or regression) or further processed by a readout layer. The model parameters $\mathcal{H}$ are optimized to minimize a task-specific loss (e.g., cross-

entropy for classification or mean squared error for regression tasks) over a training set. Inheriting the stability to perturbations of neural network structures (Gama et al., 2020a; Bruna & Mallat, 2013), VNNs have been shown stable to errors in covariance estimation, as stated in the following theorem.

**Theorem 1** ((Sihag et al., 2022)). *Consider a VNN $\Phi(\mathbf{x}, \mathbf{C}, \mathcal{H})$ of $L$ layers with $F_l = F \quad \forall l$ and nonlinearities $\sigma(\cdot)$ such that $|\sigma(a) - \sigma(b)| \leq |a - b|$. Let $\mathbf{H}(\mathbf{C})$ denote a generic covariance filter in the VNN and let each filter be Lipschitz with constant $P$ (cf. Def. 1). Consider a generic data sample $\mathbf{x}$ with covariance $\mathbf{C}$ and $\|\mathbf{x}\| \leq 1$ w.l.o.g.. Then, with probability $1 - o(1)$ it holds that*

$$\|\mathbf{H}(\hat{\mathbf{C}}) - \mathbf{H}(\mathbf{C})\| \leq \frac{P}{\sqrt{t}} \mathcal{O}\left(\sqrt{N} + \frac{\|\mathbf{C}\|\sqrt{\log(Nt)}}{\nu t}\right) = \beta \text{ and } \|\Phi(\mathbf{x}, \mathbf{C}, \mathcal{H}) - \Phi(\mathbf{x}, \hat{\mathbf{C}}, \mathcal{H})\| \leq L F^{L-1}\beta$$

*for some $\nu > 0$ capturing the data distribution, and $\|\cdot\|$ denoting the $2-$ or the spectral norm.*

We refer to (Sihag et al., 2022) for details on the probability which tends to 1 as $t$ increases and on $\nu$ which is not part of our following analysis. This notion of stability follows an extensive line of research on stability of GNNs to generic graph perturbations (Gama et al., 2020b; Cerviño et al., 2022; Arghal et al., 2022; Parada-Mayorga & Ribeiro, 2021; Keriven et al., 2020; Ruiz et al., 2020), but it specifically considers covariance estimation errors, which is fundamental for VNNs. Indeed, since in practical applications the true covariance is not available and VNNs operate on an estimate of it, this notion of stability acts as a certificate of performance guarantee w.r.t. the ideal scenario when working with the true covariance matrix and characterizes the impact of data and model characteristics on the performance. More in detail, the output difference of VNN caused by the sample estimation error is bounded proportionally by the square root of the number of samples $t$. When $t$ increases, the bound approaches zero, ultimately, showing convergence to the true data distribution. This result is specific to covariance estimation errors, whereas generic GNN stability bounds do not decrease with the number of samples. This improves the stability w.r.t. PCA, which we provide in the following lemma.

**Lemma 1.** *Consider a true covariance matrix $\mathbf{C}$ and a sample covariance estimate $\hat{\mathbf{C}}$ with respective eigendecompositions $\mathbf{C} = \mathbf{V}\mathbf{\Lambda}\mathbf{V}^\mathsf{T}$ and $\hat{\mathbf{C}} = \hat{\mathbf{V}}\hat{\mathbf{\Lambda}}\hat{\mathbf{V}}^\mathsf{T}$. Then, for any signal $\mathbf{x}$ with $\|\mathbf{x}\| \leq 1$, it holds with probability $1 - o(1)$ that $\|\mathbf{V}^\mathsf{T}\mathbf{x} - \hat{\mathbf{V}}^\mathsf{T}\mathbf{x}\| \leq \mathcal{O}(t^{-1/2}(\min_{i,j\neq i}|\lambda_i - \lambda_j|)^{-1})$.*

We refer to (Vershynin, 2018) for details on the probability, which gets closer to 1 as $t$ increases. Lemma 1 shows that PCA stability is inversely proportional to the smallest gap between covariance eigenvalues, leading to unstable behaviors when the eigenvalues are close. VNNs do not suffer from this as the covariance filter can exhibit a stable response to close eigenvalues at the expense of lower discriminability (Sihag et al., 2022), which is modeled by their frequency response and Lipschitz constant $P$. Because of this advantage in finite-data settings, VNNs have been shown effective in covariance-based learning tasks, both on static and temporal data (Sihag et al., 2024; 2023; Cavallo et al., 2024b;a). However, they have two major limitations. First, when the true covariance matrix is sparse, the finite-sample estimate contains spurious correlations (Baik et al., 2005; Paul, 2007), affecting the performance of the VNN significantly. Second, when the true covariance is dense, a VNN on the finite-sample estimate is limited by its high quadratic computational complexity in the data dimension, restricting its applicability to low-dimensional settings only.

**Paper objective.** Our goal is to overcome the above limitations by sparsifying the sample covariance matrix for the VNN. While PCA-based processing sparsification has been studied (Bickel & Levina, 2008b; Deshpande & Montanari, 2016), its extension to the VNN is challenging because it is unclear: (i) how to perform sparsification depending on whether the true covariance matrix is sparse or dense; (ii) what are the effects of sparsification on VNN stability. When the true covariance matrix is sparse, we apply thresholding-based sparsification strategies for the sample covariance matrix, and prove that they result in stability improvements (Thms. 2-3). When the true covariance matrix is dense, we put forth a stochastic sparsification framework in a form akin to dropout (Def. 4), characterize its impact on the VNN stability (Thm. 4), and propose principled sparsification strategies based on these findings (Sec. 4.3). All proofs are collected in the appendix.

## 3 SPARSE TRUE COVARIANCE

In applications involving brain and spectroscopic imaging, weather forecasting, and finance, the correlations between data points are generally sparse (e.g., only some brain regions activate simulta-

neously, only some stock prices are affected by similar factors, etc.). However, the sample covariance matrix is notoriously prone to spurious correlations due to limited sample size (Bickel & Levina, 2008a; Jobson & Korkie, 1980; Ledoit & Wolf, 2003), and sparsity-based covariance regularizers have been proposed as a backup (Bickel & Levina, 2008b). In this section, we consider sparse true covariance matrices, propose two sparsification strategies, hard and soft thresholding, and analyze their effects on the VNN embeddings. The choice of hard and soft thresholding instead of other regularized covariance estimations (Ledoit & Wolf, 2003; Bickel & Levina, 2008a; Bien & Tibshirani, 2011; Friedman et al., 2007) lies in their computational efficiency and theoretical tractability.

We now define the frequency response of a covariance filter, which will be instrumental for our analysis. By computing the graph Fourier transform of input and output signal of the covariance filter (cf. equation 1) we get $\tilde{\mathbf{u}} = \hat{\mathbf{V}}^\mathsf{T}\mathbf{u} = \hat{\mathbf{V}}^\mathsf{T}\sum_{k=0}^{K} h_k[\hat{\mathbf{V}}\hat{\mathbf{\Lambda}}\hat{\mathbf{V}}^\mathsf{T}]^k\mathbf{x} = \sum_{k=0}^{K} h_k\hat{\mathbf{\Lambda}}^k\hat{\mathbf{V}}^\mathsf{T}\mathbf{x}$, which, for the $i$-th entry, leads to $\tilde{\mathbf{u}}_i = \sum_{k=0}^{K} h_k\hat{\lambda}_i^k\tilde{\mathbf{x}}_i = h(\hat{\lambda}_i)\tilde{\mathbf{x}}_i$. That is, the frequency response of the covariance filter is a polynomial $h(\lambda) = \sum_{k=0}^{K} h_k\lambda^k$ with frequency variable $\lambda$ specified on the eigenvalues $\hat{\lambda}_i$.

**Definition 1** (Lipschitz covariance filter). *The covariance filter is Lipschitz with constant $P$ if, for every pair of eigenvalues $\lambda_i, \lambda_j \in [0, \lambda_{\max}]$, $\lambda_i \neq \lambda_j$, its frequency response satisfies: $|h(\lambda_i) - h(\lambda_j)| \leq P|\lambda_i - \lambda_j|$, where $\lambda_{\max} \in [0, \infty)$ identifies a suitable range for the covariance eigenvalues.*

Lipschitz covariance filters control the variability of the frequency response through the constant $P$. Higher $P$ allows the frequency response to generate different outputs at close eigenvalues and improves the filter's discriminability, but may degrade its stability as shown in the following.

### 3.1 HARD THRESHOLDING

Hard thresholding removes the covariance entries below a given value and it has been shown to improve covariance estimation in high-dimensional low-data settings (Bickel & Levina, 2008b).

**Definition 2** (Hard thresholding). *Given the sample covariance matrix $\hat{\mathbf{C}}$ and a coefficient $\tau > 0$, the hard thresholding function is $\eta(\hat{\mathbf{C}})_{ij} = \hat{c}_{ij}$ if $|\hat{c}_{ij}| \geq \tau/\sqrt{t}$, 0 otherwise.*

The hard threshold is inversely dependent on the number of samples as $1/\sqrt{t}$. This follows the intuition that the non-thresholded estimator approaches the true covariance as $t$ increases, hence, the sparsification is less needed and disappears in the limit of $t \to \infty$. Hard thresholding provides a more reliable covariance estimate because it removes small spurious finite-sample errors and, thus, improves the performance of VNNs when the true covariance is sparse. We analyze the impact of hard thresholding on the VNN stability in the following theorem.

**Theorem 2.** *Let the true covariance $\mathbf{C}$ belong to the sparse class $\mathcal{C} = \{\mathbf{C} : c_{ii} \leq M, \sum_{j=1}^{N} \mathbb{1}[c_{ij} \neq 0] \leq c_0, \forall i\}$ where $M > 0$ is a constant, $\mathbb{1}(\cdot)$ is the indicator function and $c_0$ is the maximum number of non-zero elements in each row of $\mathbf{C}$. Consider a hard-thresholded sample covariance matrix $\bar{\mathbf{C}}$ following Def. 2 with $\tau = M'\sqrt{\log N}$ and $M'$ large enough. With probability $1 - o(1)$, it holds that*

$$\|\mathbf{H}(\bar{\mathbf{C}})\mathbf{x} - \mathbf{H}(\mathbf{C})\mathbf{x}\| \leq t^{-1/2}Pc_0\sqrt{N\log N}(1 + \sqrt{2N}) + \mathcal{O}\left(t^{-1}\right).$$

We refer to (Bickel & Levina, 2008b) for details on the probability, which gets closer to 1 as the number of samples increases. Thm. 2 shows that hard-thresholded covariance filters –and consequently VNN from the results in Thm. 1– are stable to covariance estimation errors and that they converge to the respective filter and VNN operating with the true covariance as $1/\sqrt{t}$ in the number of data samples. Importantly, hard thresholding provides a tighter stability bound than dense VNNs (Sihag et al., 2022) and sparse PCA (Bickel & Levina, 2008b).

*Comparison with dense VNN.* Contrasting the results of Thm. 2 and Thm. 1, we see that both bounds decrease with the same order $\mathcal{O}(t^{-1/2})$. However, the hard-thresholded VNN bound depends linearly on the number of non-zero elements in a row $c_0$, whereas that of the dense VNN depends on the spectral norm $\|\mathbf{C}\|$. For sparse covariance, $c_0 \ll \|\mathbf{C}\|$ and the stability bound is tighter.

*Comparison with sparse PCA.* To provide further insights on the stability of sparse covariance filters, we contrast it also with the stability of sparse PCA.

**Proposition 1.** *Consider a true covariance matrix $\mathbf{C}$ and the thresholded sample covariance estimate $\bar{\mathbf{C}}$ with respective eigendecompositions $\mathbf{C} = \mathbf{V}\mathbf{\Lambda}\mathbf{V}^\mathsf{T}$ and $\bar{\mathbf{C}} = \bar{\mathbf{V}}\bar{\mathbf{\Lambda}}\bar{\mathbf{V}}^\mathsf{T}$. Then, for any signal $\mathbf{x}$ with*

$\|\mathbf{x}\| \leq 1$, *it holds with probability* $1 - o(1)$ *that*

$$\|\mathbf{V}^\top \mathbf{x} - \bar{\mathbf{V}}^\top \mathbf{x}\| \leq t^{-1/2} (\min_i |\lambda_i - \lambda_{i+1}|)^{-1} c_0 N \sqrt{2 \log N} \tag{2}$$

That is, the stability of sparse PCA is inversely proportional to the minimum eigenvalue gap. This term is not present in the stability of S-VNN, i.e., Thm. 2, as it is absorbed by the filter Lipschitz constant $P$: indeed, the covariance filter can exhibit a stable frequency behavior for close eigenvalues at the expense of lower discriminability, but the latter is compensated in the subsequent VNN layers that use cascades of filterbanks and non-linearities. Thus, hard-thresholded VNNs attain improved stability w.r.t. sparse PCA in sparse covariance setting.

### 3.2 SOFT THRESHOLDING

When the data follows the spiked covariance model, such as in electrocardiogram or brain image data (Johnstone & Lu, 2009), soft thresholding has been studied as a better alternative to achieve more reliable covariance estimates (Deshpande & Montanari, 2016). In this model, data points follow $\mathbf{x}_i = \sum_{q=1}^{r} \sqrt{\beta_q} u_{q,i} \mathbf{v}_q + \mathbf{z}_i$, where $\mathbf{v}_1, \ldots, \mathbf{v}_r \in \mathbb{R}^N$ are orthonormal vectors with exactly $c_0$ non-zero entries of magnitudes lower-bounded by $\theta / \sqrt{c_0}$ for some constant $\theta > 0$, $u_{q,i} \sim \mathcal{N}(0,1)$ and $\mathbf{z}_i \sim \mathcal{N}(\mathbf{0}, \mathbf{I})$ are i.i.d., and $\beta_q \in \mathbb{R}_+$ is a measure of the signal-to-noise ratio. In this case, we sparsify the covariance estimate with soft thresholding.

**Definition 3** (Soft thresholding). *Given the sample covariance matrix* $\hat{\mathbf{C}}$ *and a coefficient* $\tau > 0$, *we define the soft thresholding function as* $\eta(\hat{\mathbf{C}})_{ij} = \hat{c}_{ij} - \text{sign}(\hat{c}_{ij}) \tau / \sqrt{t}$ *if* $|\hat{c}_{ij}| > \tau / \sqrt{t}$, *0 otherwise.*

Unlike hard thresholding which only removes small noisy values, soft thresholding subtracts a value from all entries to remove noise also from non-zeroed coefficients. Again, we set the threshold to decrease with the number of samples analogously to (Deshpande & Montanari, 2016) as more accurate covariance estimates reduce the need for sparsification. The following theorem links soft thresholding to the VNN stability.

**Theorem 3.** *Consider a soft-thresholded estimate of the covariance matrix* $\bar{\mathbf{C}}$ *as per Def. 3 with* $\tau = M' \sqrt{\log(N/c_0^2)}$ *and* $M', C$ *two large enough constants. Let the eigenvalues of the true covariance* $\{\lambda_i\}_{i=0}^{N-1}$ *be all distinct. Then, the following holds with probability* $1 - o(1)$:

$$\|\mathbf{H}(\bar{\mathbf{C}})\mathbf{x} - \mathbf{H}(\mathbf{C})\mathbf{x}\| \leq t^{-1/2} P \sqrt{N} C c_0 \max(1, \lambda_{\max}) \sqrt{\max(\log(N/c_0^2), 1)} (1 + \sqrt{2N}) + \mathcal{O}(t^{-1}).$$

We refer to (Deshpande & Montanari, 2016) for details on the probability, which gets closer to 1 as $t$ increases. Thm. 3 shows that VNNs are stable also when the covariance estimate is soft thresholded and the stability bound decreases with the number of samples as $t^{-1/2}$. The main takeaways for hard thresholding w.r.t. dense VNNs and sparse PCA hold also for soft-thresholded VNNs. Furthermore, soft thresholding provides better stability than hard thresholding in low-data settings under spiked covariance, as the bound in Thm. 3 contains $\sqrt{\log(N/c_0^2)}$ which is smaller than $\sqrt{\log N}$ in Thm. 2.

**Computational complexity.** A sparse VNN layer has a computational complexity of order $\mathcal{O}(\|\bar{\mathbf{C}}\|_0 K F_{\text{in}} F_{\text{out}})$, where $\|\bar{\mathbf{C}}\|_0$ is the number of non-zero values of the thresholded sample covariance. This is significantly better than a VNN with dense covariance matrix, which is of order $\mathcal{O}(N^2 K F_{\text{in}} F_{\text{out}})$, since $\|\bar{\mathbf{C}}\|_0$ is generally much smaller than $N^2$.

## 4 GENERIC TRUE COVARIANCE

Often, the true covariance matrix and its estimate are both dense which makes the VNN computationally heavy. While we can use hard or soft thresholding or any graph sparsification approach, the resulting sparse VNN may be significantly different from the VNN operating on the true covariance, ultimately, leading to divergent outputs and degraded performance. To reduce the computational cost in a tractable manner, we propose a stochastic sparsification framework in a form akin to dropout. Such an approach is general and does not have any structure assumption on the covariance matrix.

**Definition 4** (Stochastic sparsification). *Let* $\boldsymbol{\Delta}$ *be a matrix of the same support as the sample covariance* $\hat{\mathbf{C}}$ *with entries* $\delta_{ij} = \delta_{ji} = 1$ *with probability* $p_{ij}$ *and 0 otherwise (i.e.,* Bernoulli$(p_{ij})$*) and* $\delta_{ii} = 1$. *A sparsified covariance matrix is* $\tilde{\mathbf{C}} = \boldsymbol{\Delta} \odot \hat{\mathbf{C}}$, *where* $\odot$ *is the elementwise product.*

Stochastic sparsification generates random sparsified matrices $\tilde{\mathbf{C}}$ of the sample covariance matrix via element-wise independent sampling in a way that $\tilde{\mathbf{C}}$ preserves: (i) the symmetric property in consistency with the covariance principle; and (ii) the variance of the data points on the main diagonal. The sparsification can be controlled by the probabilities $p_{ij}$ depending on application requirements.

## 4.1 VNNs with stochastic covariance sparsification

We now investigate the effects of stochastic sparsification on VNNs' stability by framing the stochastic sparsification as a stochastic perturbation problem following (Gao et al., 2021a) and defining stochastic sparsified covariance filters.

**Definition 5** (Stochastic covariance filter). *Given a sequence of i.i.d. realizations $\tilde{\mathbf{C}}_k, \ldots, \tilde{\mathbf{C}}_1$ of the stochastic sparsified covariance $\tilde{\mathbf{C}}$ independent on data distribution, a covariance filter $\mathbf{H}(\tilde{\mathbf{C}})$ performs convolution of a generic signal $\mathbf{x}$ as $\tilde{\mathbf{u}} = \mathbf{H}(\tilde{\mathbf{C}})\mathbf{x} = \sum_{k=0}^{K} h_k \tilde{\mathbf{C}}_k \ldots \tilde{\mathbf{C}}_1 \tilde{\mathbf{C}}_0 \mathbf{x}$ with $\tilde{\mathbf{C}}_0 = \mathbf{I}$.*

Stochastic filters shift the signal $\mathbf{x}$ over $K$ different random realizations $\{\tilde{\mathbf{C}}_k\}_{k=1}^{K}$ of $\tilde{\mathbf{C}}$ rather than shifting over a fixed $\hat{\mathbf{C}}$. This leads to a generalized frequency interpretation.

**Definition 6** (Generalized covariance filter frequency response). *The generalized frequency response of the stochastic covariance filter is a multivariate function of the*

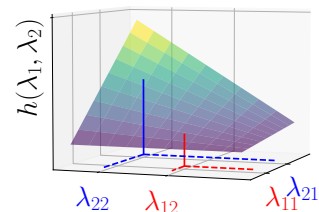

Figure 1: Two-dimensional generalized frequency response $h(\boldsymbol{\lambda})$ of stochastic sparsified covariance filter. $h(\boldsymbol{\lambda})$ is determined by the coefficients $\{h_k\}_{k=0}^{K}$ and is independent of the covariance realizations. For specific covariance realizations, $h(\boldsymbol{\lambda})$ is instantiated on the corresponding multivariate frequencies (e.g., $\lambda_{11}, \lambda_{12}$ and $\lambda_{21}, \lambda_{22}$ for two different realizations $\tilde{\mathbf{C}}_1\tilde{\mathbf{C}}_2$ and $\tilde{\mathbf{C}}_1'\tilde{\mathbf{C}}_2'$).

*form $h(\boldsymbol{\lambda}) = \sum_{k=0}^{K} h_k \prod_{\kappa=0}^{k} \lambda_\kappa$ where $\boldsymbol{\lambda} = [\lambda_1, \ldots, \lambda_K]^\mathsf{T}$ with $\lambda_0 = 1$ is a generic vector variable and each frequency variable $\lambda_k$ corresponds to the covariance realization $\tilde{\mathbf{C}}_k$ for $k = 1, \ldots, K$.*

The derivations of such a frequency response are reported in Appendix E. Fig. 1 illustrates an example of $K = 2$. This extends the deterministic case, in which all realizations and corresponding eigenvalues are equal with $\prod_{\kappa=0}^{k} \lambda_\kappa = \lambda^k$. Given the generalized frequency response, we generalize the integral Lipschitz property for stability analysis.

**Definition 7** (Lipschitz gradient). *Consider the analytic generalized frequency response $h(\boldsymbol{\lambda})$ and two instantiations $\boldsymbol{\lambda}_1 = [\lambda_{11}, \ldots, \lambda_{1K}]^\mathsf{T}$ and $\boldsymbol{\lambda}_2 = [\lambda_{21}, \ldots, \lambda_{2K}]^\mathsf{T}$ of vector variable $\boldsymbol{\lambda}$. Consider also the auxiliary vector that concatenates the first $k$ entries of $\boldsymbol{\lambda}_2$ and the last $K - k$ entries of $\boldsymbol{\lambda}_1$, i.e., $\boldsymbol{\lambda}^{(k)} = [\lambda_{21}, \ldots, \lambda_{2k}, \lambda_{1(k+1)}, \ldots, \lambda_{1K}]^\mathsf{T}$. The Lipschitz gradient of $h(\boldsymbol{\lambda})$ between $\boldsymbol{\lambda}_1$ and $\boldsymbol{\lambda}_2$ is*

$$\nabla_L h(\boldsymbol{\lambda}_1, \boldsymbol{\lambda}_2) = \left[ \partial h(\boldsymbol{\lambda}^{(1)})/\partial \lambda_1, \ldots, \partial h(\boldsymbol{\lambda}^{(K)})/\partial \lambda_K \right]^\mathsf{T}$$

*where $\partial h(\boldsymbol{\lambda}^{(k)})/\partial \lambda_k$ is the partial derivative w.r.t. $\lambda_k$ at $\boldsymbol{\lambda}^{(k)}$.*

**Definition 8** (Generalized integral Lipschitz filter). *A covariance filter is generalized integral Lipschitz if there exists a constant $P$ s.t. $\|\nabla_L h(\boldsymbol{\lambda}_1, \boldsymbol{\lambda}_2)\|_2 \leq P$ and $\|\boldsymbol{\lambda}_1 \odot \nabla_L h(\boldsymbol{\lambda}_1, \boldsymbol{\lambda}_2)\|_2 \leq P$.*

The Lipschitz gradient characterizes the variability of $h(\boldsymbol{\lambda})$ since, for two multivariate frequency vectors $\boldsymbol{\lambda}_1, \boldsymbol{\lambda}_2$, we have $h(\boldsymbol{\lambda}_2) - h(\boldsymbol{\lambda}_1) = \nabla_L^\mathsf{T} h(\boldsymbol{\lambda}_1, \boldsymbol{\lambda}_2)(\boldsymbol{\lambda}_2 - \boldsymbol{\lambda}_1)$. The generalized integral Lipschitz filter limits this variability to be at most linear in the multidimensional space and to decrease as the frequency $\boldsymbol{\lambda}$ is specified at large values, which extends the standard integral Lipschitz property to the multivariate frequency domain.

## 4.2 Stability of VNNs with stochastic covariance sparsification

These preliminaries allow us to analyze the stability of VNNs to stochastic covariance sparsification.

**Theorem 4.** *Consider a randomly sparsified covariance filter $\mathbf{H}(\tilde{\mathbf{C}})$ [cf. Def. 5] that is generalized integral Lipschitz with constant $P$ [cf. Def. 8]. Let also $\mathbf{H}(\mathbf{C})$ denote the same filter (fixed parameters) operating on the true covariance matrix. Then, for a generic signal $\|\mathbf{x}\| \leq 1$, the expected squared*

*difference between the two filters can be upper-bounded as*

$$\mathbb{E}[\|\mathbf{H}(\mathbf{C})\mathbf{x} - \mathbf{H}(\tilde{\mathbf{C}})\mathbf{x}\|^2] \leq \underbrace{NP^2Q + \mathcal{O}((1-p_1)(1-p_2))}_{\text{sparsification error}} + \underbrace{\frac{P^2}{t}\mathcal{O}\left(N + \frac{\|\mathbf{C}\|^2\log(Nt)}{\nu^2 t^2}\right)}_{\text{covariance uncertainty}}$$

*where $Q = \sum_{i=1}^{N}\sum_{n=1}^{N}\hat{c}_{in}^2(1-p_{in})$, $p_1, p_2$ are generic probabilities such that $\mathcal{O}((1-p_1)(1-p_2))$ is dominated by the linear terms in the probability value in $Q$ and $\nu$ is defined in Thm. 1.*

Thm. 4 identifies two main factors that affect the stability of stochastic sparsified covariance filters: the covariance uncertainty and the sparsification error.

*Covariance uncertainty.* This term is analogous to the dense case in Thm. 1. It decreases with the number of samples as $\mathcal{O}(1/t)$ and is dominated by the sparsification error for sufficiently large $t$. Note that a large $t$ does not necessarily correspond to a small covariance estimation error here because the stochastic sparsification may still remove large covariance values; ultimately, making our analysis principally different from (Sihag et al., 2022) which assumes small covariance perturbations.

*Sparsification error.* This term decreases as the sampling probabilities $p_{ij} \to 1$, corresponding to an improved stability but a lower sparsification. This indicates a trade-off between the perturbation effect caused by the covariance sparsification error and the computational cost saved by the stochastic sparsification. The stability constant depends on the filter Lipschitz constant $P$, the data dimension $N$, and the coupling between the covariance values $\hat{c}_{ij}$ and the sampling probabilities $p_{ij}$, i.e., $Q$. A larger $P$ allows for a higher filter discriminability as the frequency response can change more quickly but leads to worse stability. Data with high dimensionality $N$ results in a larger graph and increases the effect of sparsification on the VNN stability. More importantly, $Q$ represents the interplay between sample covariance values $\hat{c}_{ij}$ and their corresponding sampling probabilities $p_{ij}$, which can be used as a design choice to develop different stochastic sparsification strategies in a principled manner as we shall discuss in the next section. Finally, note that the results in Thm. 4 and its proof generalize and differ substantially from the results in (Gao et al., 2021a, Theorem 1) as we consider here edge-specific probabilities $p_{ij}$ rather than an identical probability $p_{ij} = p$ and we identify a connection between VNN stability and the data distribution through their covariances $\hat{c}_{ij}$, which is not present in (Gao et al., 2021a).

### 4.3 Stochastic sparsification strategies

Following our theoretical analysis, we propose two strategies to assign the sparsification probabilities to covariance values and assess their impact on the stability of VNN.

**Absolute covariance values (ACV).** Despite being dense, the covariance matrix may contain some values of lower magnitude due to spurious correlations. We want to drop these small values with a high probability to save computational cost while keeping large covariance values to maintain the useful dependencies. Thus, we define the probability as $p_{ij} = |\hat{c}_{ij}|/\hat{c}_{\max}$, where $\hat{c}_{\max} = \max_{i,j}|\hat{c}_{ij}|$. Fig. 2 (left) shows that the stability term $q_{in}$ within $Q$ [cf. Thm. 4] is small when the covariance value $\hat{c}_{in}$ approaches zero or the maximal value $\hat{c}_{\max}$. This is because small covariance values have little impact on the VNN stability even if they are more likely to be dropped, and large covariance values (close to $\hat{c}_{\max}$) are less likely to be dropped though their removal would affect stability more.

**Ranked covariance values (RCV).** While ACV improves efficiency by preserving stronger correlations, it does not allow any control over the amount of sparsification (e.g., if all covariance values are high, very few are dropped). To overcome this, we define a set of probability values with a desired mean $p$ and assign them to covariance values based on their positions in this absolute ranking. Formally, consider an ordered set of probabilities $\mathcal{P} = \{p_1', \ldots, p_{N'}'\}$, where $p_i' \leq p_{i+1}'$, and each $p_i'$ is sampled from $\mathcal{N}(p, \sigma)$ with $\sigma = \min((1-p)/3, p/3))$ (such that the number of values not in $[0, 1]$, which we clip to the interval, is negligible) and $N'$ is the number of probability values for assignment. We set $p_{ij} = p_k'$ where $k = |\{c_{lm} : |c_{lm}| < |c_{ij}|; l, m = 1, \ldots, N\}|$ is the position of $c_{ij}$ in the ordered ranking of absolute covariances. Consequently, the expected percentage of dropped covariances is $1 - p$, which allows controlling the sparsification level at the risk of removing useful covariances. In practice, the value of $p$ can be treated as a hyperparameter and tuned through cross-validation or on a validation set based on the desired accuracy and stability.

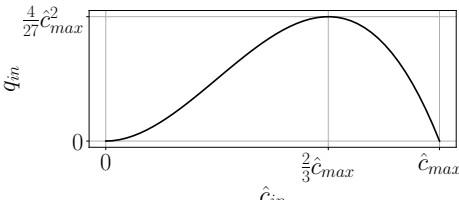 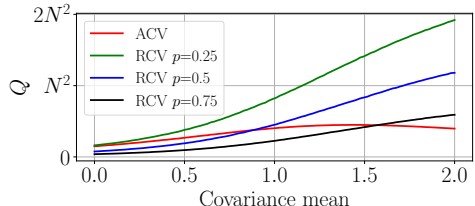

Figure 2: Insights into stability for stochastic sparsification. (Left) Stability bound elements $q_{in} = \hat{c}_{in}^2 (1 - |\hat{c}_{in}|/\hat{c}_{\max})$ with $Q = \sum_{i=1}^N \sum_{n=1}^N q_{in}$ for ACV [cf. Thm. 4]. (Right) Stability term $Q$ [cf. Thm. 4] for covariance values distributions with different means and stochastic sparsifications.

Fig. 2 (right) shows how the term $Q$ changes for ACV and RCV with different means of covariance value distribution. RCV with a smaller $p$ corresponds to lower stability especially when the covariance values are high, because they are dropped regardless of their value. ACV, instead, maintains a consistent level of stability since it balances the covariance magnitudes and the dropping probabilities.

**Computational complexity.** The expected computational complexity of a VNN layer with stochastic sparsification is of the order $\mathcal{O}(\mathbb{E}[\|\tilde{\mathbf{C}}\|_0] K F_{\text{in}} F_{\text{out}})$, where $\mathbb{E}[\|\tilde{\mathbf{C}}\|_0]$ is the expected number of non-zero elements of the sparsified covariance. For ACV, we have that $\mathbb{E}[\|\tilde{\mathbf{C}}\|_0] = \hat{c}_{\text{mean}}(\|\hat{\mathbf{C}}\|_0 - N) + N$, where $\hat{c}_{\text{mean}} = 1/N^2 \sum_{i,j=1,\ldots,N, i \neq j} |\hat{c}_{ij}|$, so the complexity depends on the data. Instead, for the RCV we have $\mathbb{E}[\|\tilde{\mathbf{C}}\|_0] = p(\|\hat{\mathbf{C}}\|_0 - N) + N$, so the complexity can be controlled through $p$.

**Remark 1.** *The stochastic sparsification with RCV generalizes the hard thresholding approach in Sec. 3, which can be recovered by setting probabilities as $p_{ij} = 1$ if $|c_{ij}| > \tau/\sqrt{t}$, 0 otherwise.*

**Remark 2.** *Both thresholding and stochastic sparsification provide symmetric covariance estimates but do not necessarily preserve positive semidefiniteness (PSD), which is common in related works (Bickel & Levina, 2008b; Deshpande & Montanari, 2016; Li et al., 2021; Liao et al., 2022) and places our techniques within the scope of the literature. Bickel & Levina (2008b) also provide a sufficient condition for PSD under hard-thresholding, which we report in Appendix G.*

## 5 NUMERICAL RESULTS

We corroborate the proposed methods with experiments on real and synthetic datasets, targeting the following objectives: validate our theoretical results on (**O1**) thresholding when the true covariance matrix is sparse and (**O2**) stochastic sparsification for a generic true covariance; (**O3**) compare S-VNNs and VNNs on real datasets. We provide details and additional results in Appendix I.

### 5.1 STABILITY OF SPARSE VNN

**Experimental setup – (O1)-(O2).** We generate synthetic data with a controlled covariance by sampling data points $\mathbf{x}_i \sim \mathcal{N}(\mathbf{0}, \mathbf{C})$. Then, we create regression targets as $y_i = \mathbf{w}^\mathsf{T} \mathbf{x}_i + u$, where $\mathbf{w}$ is a vector with elements $w_j \sim \text{Uniform}(0, 1)$ and $u \sim \mathcal{N}(0, 3)$ is a noise term. We generate 1000 samples of size $N = 100$ and divide the data into train/validation/test splits of size 80%/10%/10%.

**Sparse true covariance – (O1).** We evaluate the stability of S-VNN on a synthetic dataset with sparse covariance (*SparseCov*). We train a VNN with the true covariance and test it with different covariance estimates obtained with a varying number of samples. We compare the original sample covariance, hard thresholding (Hthr), and soft thresholding (Sthr). We use PCA-SVM as baseline, i.e., we transform data with PCA followed by an RBF-SVM for regression. Fig. 3 (left) reports the MAE w.r.t. the number of samples for covariance estimation. Overall, hard and soft thresholding provide better covariance estimates, ultimately, improving the performance (i.e., lower MAE) and increasing the stability compared to the nominal VNN. VNNs are more stable than PCA-based models, because their performance is less affected by covariance perturbations as analyzed in (Sihag et al., 2022). These results support our theoretical discussion in Sec. 3, i.e., thresholding allows VNNs to maintain stability in sparse covariance settings.

**Generic true covariance – (O2).** We assess the stochastic sparsification using synthetic datasets with large covariance values (*LargeCov*) and with small covariance values (the synthetic linear regression

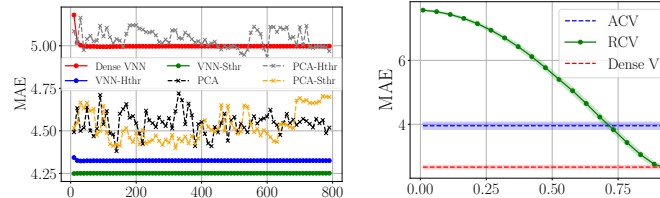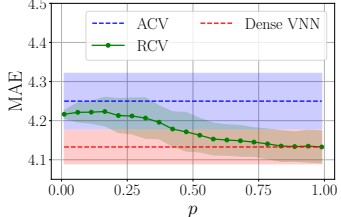

Figure 3: Stability of VNN. (Left) MAE with hard and soft thresholding on SparseCov (standard deviations are of order at most $10^{-2}$ for VNNs and $10^{-1}$ for PCA). (Center) MAE for stochastic sparsification on LargeCov. (Right) MAE for stochastic sparsification on SmallCov.

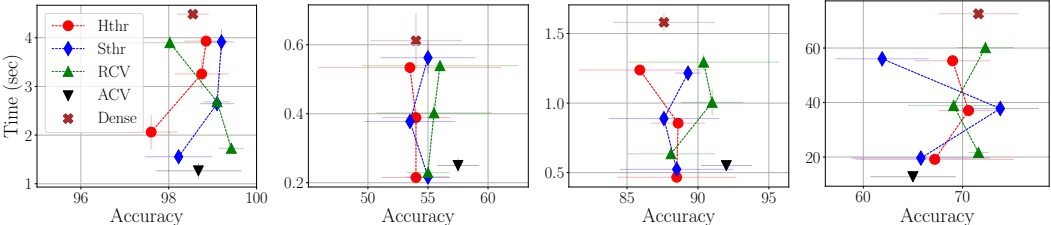

Figure 4: Accuracy and time for a forward pass for sparse and dense VNNs. From left to right, the results are for Epilepsy, CNI, MHEALTH and Realdisp. RCV results are for $p = 0.75, 0.5, 0.25$ from top to bottom, and for Hthr and Sthr we report results for 3 different thresholds to achieve analogous sparsification. Legend is shared.

dataset in (Sihag et al., 2022, Appendix E.4) with tail=0.2, here called *SmallCov*); see Appendix I.1 for the distributions of the covariance values. We train a VNN with the true covariance matrix and test it with the stochastically sparsified versions. Fig. 3 (center) and (right) compare the ACV and RCV with different values of $p$. For *LargeCov*, we see that a high sparsification leads to greater instability (RCV for small $p$) as large covariance values are dropped which hinders performance. For *SmallCov*, instead, all sparsification approaches affect stability only lightly as removing small covariance values does not lead to performance changes. ACV remains stable because it sparsifies little even when the covariance values are large. In turn, this corroborates our theoretical observations in Sec. 4.1.

## 5.2 SPARSE VNN ON REAL DATA

**Experimental setup – (O3).** We consider four real datasets of brain measurements and human action recognition data. *Epilepsy* (Kramer et al., 2008) consists of an electrocardiography time series equally split before and after an epileptic seizure event. We perform binary classification to predict whether a time sample was recorded before or after the seizure. *CNI* (Schirmer et al., 2021) contains resting-state fMRI time series of patients with attention deficit hyperactivity disorder (ADHD) and neurotypical controls (NC), which we use as labels for binary classification. *MHEALTH* (Banos et al., 2014a) and *Realdisp* (Banos et al., 2014b) contain measurements of wearable devices placed on subjects performing different actions. The goal is to classify the action performed by each subject. Additional details are reported in Appendix I.1. We compare various sparsification approaches (thresholding, RCV, ACV) against the dense VNN. For completeness, we report additional results for various PCA-based classifiers and GraphSAGE in Appendix I.3, although our objective is not to achieve state-of-the-art performance on these datasets but to reduce the impact of spurious correlations while improving computational efficiency of VNNs. For Hthr and Sthr, we set the threshold to achieve a level of sparsification comparable to RCV with $p = 0.75, 0.5, 0.25$ to ease the comparison.

**Results – (O3).** From Fig. 4, we see that the S-VNNs improve substantially on the computation time and increase the accuracy w.r.t. the nominal VNN. This is particularly emphasized for Epilepsy, CNI, and MHEALTH, where most S-VNNs perform better than dense VNN, likely because the sample covariance matrices in these datasets contain spurious correlations. On Realdisp, the accuracy improvement is smaller, indicating that in this dataset most correlations carry useful information to solve the task. Both for thresholding and RCV, changing the value of the threshold or $p$ does not lead to large changes in accuracy, meaning that the majority of covariance values are not relevant for performance, while it affects the time efficiency of the model by increasing sparsity. Finally, on 3 out

of 4 datasets, stochastic sparsification achieves a slightly superior performance than deterministic thresholding, likely due to the regularization effect of different realizations of a sparsified covariance over different runs in a form akin to dropout-like techniques (Papp et al., 2021; Rong et al., 2020).

# 6 RELATED WORKS

*Sparse PCA.* The instability of the finite-sample covariance matrix is widely studied for PCA (Jolliffe, 2002; Paul, 2007; Baik et al., 2005) and various regularized estimators have been proposed including covariance shrinkage (Ledoit & Wolf, 2003; 2012), lasso penalties (Bien & Tibshirani, 2011; Han Liu & Zhao, 2014), and thresholding (Bickel & Levina, 2008b; Deshpande & Montanari, 2016). We leverage here hard and soft thresholding given their benefits in PCA in low-data sparse-covariance settings and study their impact on VNN stability.

*Stability of GNNs-VNNs.* VNNs can be seen as graph convolutional neural networks operating on the covariance graph (Sihag et al., 2022), which draws analogies with PCA, but have been shown to be more stable extending the small perturbation analysis of GNNs (Gama et al., 2020b; Kenlay et al., 2021b;a; Levie et al., 2021; Maskey et al., 2023; Gao et al., 2023) to covariances. Due to their robustness, they have proved successful in brain data processing (Sihag et al., 2024), transferability to large-dimension data (Sihag et al., 2023), temporal settings (Cavallo et al., 2024b) and biased datasets (Cavallo et al., 2024a). All these results, however, hold for the dense covariance matrix which is suboptimal in low-data regimes and computationally heavy. Here, we propose S-VNNs and study their stability when the true covariance matrix is sparse and dense. This merits a different treatment than the perturbation assumption studied in (Sihag et al., 2022) and overcomes the limitations of dense VNNs. Our stochastically sparsified VNN draws analogies with the stability of GNNs to random link drops (Gao et al., 2021a;b) but we generalize those findings to the more challenging case where all sparsified probabilities differ rather than being identical. We also propose design strategies for the sparsified probabilities based on our theoretical analysis.

*Graph sparsification in GNNs.* Graph sparsification techniques have been used for multiple purposes in GNNs but predominantly to improve: (i) scalability in large and dense graphs (Hamilton et al., 2018; Zhang et al., 2019; Zeng et al., 2021; Peng et al., 2022; Srinivasa et al., 2020; Ye & Ji, 2021); (ii) GNN expressiveness in graph classification tasks and alleviate oversmoothing (Papp et al., 2021; Rong et al., 2020; Fang et al., 2023; Zheng et al., 2020; Morris et al., 2021); and (iii) enhance interpretability (Rathee et al., 2021; Li et al., 2023; Naber et al., 2024). Our S-VNNs aim on the one hand to improve scalability and on the other to preserve the tractability and theoretical links with covariance-based data processing. While our sparsification approaches can be applied also to weighted graphs (see Appendix H), in this work we focus on covariance sparsification to characterize theoretically how this affects stability w.r.t. covariance estimation errors and spurious correlations.

# 7 CONCLUSION

In this work, we develop sparse covariance neural networks (S-VNNs) and study the effects of sparsification techniques on covariance neural networks. We show that S-VNNs are more stable to the finite-data sample effect and more computationally efficient. In particular, when the true covariance is sparse, we propose S-VNNs with hard/soft thresholding and show that they transfer to the nominal VNN with a rate inversely proportional to the square root of the number of data points. When the true covariance is dense, we put forth a stochastic sparsified approach to reduce the computational complexity while maintaining mathematical tractability and stability. In the framework of stochastic sparsification, we propose principled strategies, i.e., ACV and RCV, that align with the theoretical observations to tune sparsification impact and computation efficiency. Experimental results on four real datasets show that the proposed S-VNNs improve substantially on the computation time and achieve competitive or better performance compared with nominal VNNs, thus validating that spurious correlations are present and damaging in real datasets. Although VNNs can be applied to several other regularized covariance estimators (Ledoit & Wolf, 2003; Bickel & Levina, 2008a; Bien & Tibshirani, 2011; Friedman et al., 2007) to reduce the impact of spurious correlations, we here focus on thresholding since, unlike other estimators, it allows to control the sparsity of the covariance matrix (fundamental for time and memory efficiency), and allows for theoretical tractability. Future work will investigate other covariance estimators and characterize the regularization effect of stochastic sparsification, which allows the model to observe multiple instances of the same covariance with potential benefits for robustness, expressiveness and reduced oversmoothing.

**Reproducibility statement.** The proofs of all theoretical results are collected in the appendix. For experimental results, we make the code available with the submission and we provide references to all the datasets in the following. The preprocessing functions for all datasets are made available in the code.

- The synthetic datasets *SparseCov*, *LargeCov* and *SmallCov* are available in the code attached to the submission.

- Realdisp can be downloaded at `https://archive.ics.uci.edu/dataset/305/realdisp+activity+recognition+dataset`.

- MHEALTH can be downloaded at `https://archive.ics.uci.edu/dataset/319/mhealth+dataset`.

- CNI can be downloaded at `http://www.brainconnectivity.net/challenge.html`.

- Epilepsy can be downloaded at `https://math.bu.edu/people/kolaczyk/datasets.html` and we provide the preprocessed version in the additional material.

We report additional experimental details for reproducibility in Appendix I.2.

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

## A  BACKGROUND ON PCA AND CONNECTION WITH VNNS

PCA is a change of basis that aims at minimizing the redundancy of the transformed data while maintaining their statistical properties of interest (Shlens, 2014; Jolliffe, 2002). PCA achieves this by projecting the data onto a space where the covariance matrix of the transformed data is diagonal, i.e., the off-diagonal elements, which characterize the redundancy of the data, are all zeros. More formally, given a dataset $\mathbf{X} \in \mathbb{R}^{N \times t}$, where each column is a data sample $\mathbf{x} \in \mathbb{R}^N$, PCA applies a transformation $\tilde{\mathbf{X}} = \mathbf{T}^\mathsf{T} \mathbf{X}$ with $\mathbf{T} \in \mathbb{R}^{N \times N}$ such that the covariance of the transformed data $\tilde{\mathbf{X}} \tilde{\mathbf{X}}^\mathsf{T}$ is diagonal. This goal is achieved by setting $\mathbf{T} = \hat{\mathbf{V}}$, where $\hat{\mathbf{C}} = \mathbf{X} \mathbf{X}^\mathsf{T} = \hat{\mathbf{V}} \hat{\mathbf{\Lambda}} \hat{\mathbf{V}}^\mathsf{T}$ is the eigendecomposition of the covariance matrix of the data, since $\tilde{\mathbf{X}} \tilde{\mathbf{X}}^\mathsf{T} = \hat{\mathbf{V}}^\mathsf{T} \mathbf{X} \mathbf{X}^\mathsf{T} \hat{\mathbf{V}} = \hat{\mathbf{V}}^\mathsf{T} (\hat{\mathbf{V}} \hat{\mathbf{\Lambda}} \hat{\mathbf{V}}^\mathsf{T}) \hat{\mathbf{V}} = \hat{\mathbf{\Lambda}}$ due to the orthonormality of the covariance eigenvectors. PCA is also commonly used for dimensionality reduction by selecting only some of the eigenvectors of the covariance matrix to define the new basis, and several techniques exist to select the number of components, such as looking at the eigenvalues of the covariance and selecting the number corresponding to the largest eigenvalue drop. PCA is equivalent to the graph Fourier transform (GFT) for a graph defined by the covariance $\hat{\mathbf{C}}$ (Gama et al., 2020c; Isufi et al., 2024). Indeed, the GFT projects data on the space defined by the eigenvectors of the graph shift operator $\mathbf{S} \in \mathbb{R}^{N \times N}$ (a matrix describing the graph structure, e.g., adjacency or Laplacian), i.e., $\tilde{\mathbf{X}} = \mathbf{U}^\mathsf{T} \mathbf{X}$ given the eigendecomposition $\mathbf{S} = \mathbf{U} \mathbf{W} \mathbf{U}^\mathsf{T}$. If we consider the graph described by the sample covariance matrix, i.e., $\mathbf{S} = \hat{\mathbf{C}}$, then PCA and GFT perform the same operation.

The connection between PCA and VNNs emerges by looking at the action of the covariance filter in the spectral domain. For a data sample $\mathbf{x}$, taking the GFT of the covariance filter leads to $\tilde{\mathbf{u}} = \hat{\mathbf{V}}^\mathsf{T} \mathbf{u} = \hat{\mathbf{V}}^\mathsf{T} \sum_{k=0}^K h_k [\hat{\mathbf{V}} \hat{\mathbf{\Lambda}} \hat{\mathbf{V}}^\mathsf{T}]^k \mathbf{x} = \sum_{k=0}^K h_k \hat{\mathbf{\Lambda}}^k \hat{\mathbf{V}}^\mathsf{T} \mathbf{x}$, equivalent to $\sum_{k=0}^K h_k \hat{\lambda}_i^k \tilde{\mathbf{x}}_i = h(\hat{\lambda}_i) \tilde{\mathbf{x}}_i$ for the $i$-th entry; that is, the filter is a polynomial in the covariance eigenvalues that modulates each component of the GFT (i.e., PCA) of the signal $\mathbf{x}$ separately (cf. Sec. 3). This shows that the covariance filter processes the principal components of the data, and there exists a set of coefficients $h_k$ such that a covariance filter performs the same operation as PCA (Sihag et al., 2022, Theorem 1).

## B  PROOF OF LEMMA 1

We provide a bound for the stability of PCA, i.e., $\|\mathbf{V}^\mathsf{T} \mathbf{x} - \hat{\mathbf{V}}^\mathsf{T} \mathbf{x}\|$. We have that

$$\left\| \mathbf{V}^\mathsf{T} \mathbf{x} - \hat{\mathbf{V}}^\mathsf{T} \mathbf{x} \right\| = \left\| \sum_{i=1}^N (\mathbf{v}_i - \hat{\mathbf{v}}_i)^\mathsf{T} x_i \right\| \leq \sum_{i=1}^N \|\mathbf{v}_i - \hat{\mathbf{v}}_i\| |x_i|$$
$$\leq \sum_{i=1}^N \|\mathbf{v}_i - \hat{\mathbf{v}}_i\| \leq N \max_i \|\mathbf{v}_i - \hat{\mathbf{v}}_i\|. \tag{3}$$

where we used triangle inequality, the fact that $\|\mathbf{x}\| \leq 1$ and the fact that $\|(\mathbf{v}_i - \hat{\mathbf{v}}_i)^\mathsf{T}\| = \|\mathbf{v}_i - \hat{\mathbf{v}}_i\|$. Using the sin-theta theorem (Stewart & guang Sun, 1990, Theorem V.3.6) (cf. equation 18), we have that

$$\max_i \|\mathbf{v}_i - \hat{\mathbf{v}}_i\| \leq \frac{\sqrt{2}\|\mathbf{E}\|}{\min_i |\lambda_i - \lambda_{i+1}|} \tag{4}$$

where $\|\mathbf{E}\| = \|\mathbf{C} - \hat{\mathbf{C}}\|$. We now leverage the result from (Vershynin, 2018, Theorem 5.6.1) which shows that $\|\mathbf{E}\| \leq \mathcal{O}(t^{-1/2})$ with probability $1 - o(1)$, and this concludes the proof. □

## C  PROOF OF THMS. 2 AND 3

### C.1  PRELIMINARIES

We begin by providing a Lemma that will be used in the following.

**Lemma 2.** *Consider the true and the thresholded sample covariance matrices and their respective eigendecompositions* $\mathbf{C} = \mathbf{V} \mathbf{\Lambda} \mathbf{V}^\mathsf{T}$ *and* $\bar{\mathbf{C}} = \bar{\mathbf{V}} \mathbf{\Lambda} \bar{\mathbf{V}}^\mathsf{T}$. *The following holds:*

$$\frac{1}{\sqrt{2}} |\mathbf{v}_j^\mathsf{T} \bar{\mathbf{v}}_i| \leq \sqrt{1 - |\mathbf{v}_j^\mathsf{T} \bar{\mathbf{v}}_j|} \quad \forall i \neq j. \tag{5}$$

*Proof.* We first observe that, due to the orthonormality of $\mathbf{V}$ and $\bar{\mathbf{V}}$, we have

$$\frac{1}{2}\|\mathbf{v}_j - \bar{\mathbf{v}}_j\|^2 = \frac{1}{2}(\mathbf{v}_j - \bar{\mathbf{v}}_j)^{\mathsf{T}}(\mathbf{v}_j - \bar{\mathbf{v}}_j) = \tag{6}$$

$$\frac{1}{2}(\mathbf{v}_j^{\mathsf{T}}\mathbf{v}_j - 2\mathbf{v}_j^{\mathsf{T}}\bar{\mathbf{v}}_j + \bar{\mathbf{v}}_j^{\mathsf{T}}\bar{\mathbf{v}}_j) = 1 - |\mathbf{v}_j^{\mathsf{T}}\bar{\mathbf{v}}_j| \tag{7}$$

where we assumed $\mathbf{v}_j^{\mathsf{T}}\bar{\mathbf{v}}_j \geq 0$. Note that this assumption does not lose generality because eigenvectors are invariant to change of sign, i.e., if $\mathbf{v}_j$ is an eigenvector of $\mathbf{C}$, $-\mathbf{v}_j$ is equivalently an eigenvector. Now, we write $\mathbf{v}_j - \bar{\mathbf{v}}_j = \delta\mathbf{v}_j$ as the sum of two components, $\delta\mathbf{v}_j = \delta\mathbf{v}_{j\|} + \delta\mathbf{v}_{j\perp}$, that are respectively parallel and perpendicular to $\mathbf{v}_j$. Due to triangle inequality, we have that $\|\delta\mathbf{v}_{j\perp}\| + \|\delta\mathbf{v}_{j\|}\| \leq \|\delta\mathbf{v}_{j\perp}\| \leq \|\delta\mathbf{v}_j\|$. Since the columns of $\mathbf{V}$ form an orthonormal basis of $\mathbb{R}^N$, the perpendicular component of $\delta\mathbf{v}_j$ is the sum of the projections of $\bar{\mathbf{v}}_j$ on all columns of $\mathbf{V}$ except $\mathbf{v}_j$, i.e., $\delta\mathbf{v}_{j\perp} = \sum_{i=1, i\neq j}^{N}(\bar{\mathbf{v}}_j^{\mathsf{T}}\mathbf{v}_i)\mathbf{v}_i$ and the norm becomes $\|\delta\mathbf{v}_{j\perp}\| = \sqrt{\sum_{i=1, i\neq j}^{N}(\bar{\mathbf{v}}_j^{\mathsf{T}}\mathbf{v}_i)^2}$. Due to triangle inequality and orthonormality of $\mathbf{V}$, for any $i \neq j$, we have

$$\frac{1}{\sqrt{2}}|\bar{\mathbf{v}}_j^{\mathsf{T}}\mathbf{v}_i| \leq \frac{1}{\sqrt{2}}\|\delta\mathbf{v}_{j\perp}\| \leq \frac{1}{\sqrt{2}}\|\mathbf{v}_j - \bar{\mathbf{v}}_j\| = \sqrt{1 - |\mathbf{v}_j^{\mathsf{T}}\bar{\mathbf{v}}_j|}. \tag{8}$$

$\square$

We now provide a general result on VNN stability that will be the starting point to prove Thms. 2 and 3.

**Proposition 2** (Stability of VNNs). *Consider a Lipschitz covariance filter $\mathbf{H}(\mathbf{C})$ with constant $P$. Let $\bar{\mathbf{C}}$ be an estimated covariance matrix of $\mathbf{C}$. Then, for any generic signal $\mathbf{x}$ where $\|\mathbf{x}\| \leq 1$ and estimation error $\mathbf{E} = \mathbf{C} - \bar{\mathbf{C}}$ where $\|\mathbf{E}\| \ll 1$, it holds that*

$$\|\mathbf{H}(\bar{\mathbf{C}})\mathbf{x} - \mathbf{H}(\mathbf{C})\mathbf{x}\| \leq P\sqrt{N}\|\mathbf{E}\|(1 + \sqrt{2N}) + \mathcal{O}(\|\mathbf{E}\|^2). \tag{9}$$

*Proof.* We will make use of the eigendecompositions $\mathbf{C} = \mathbf{V}\mathbf{\Lambda}\mathbf{V}^{\mathsf{T}}$ and $\bar{\mathbf{C}} = \bar{\mathbf{V}}\bar{\mathbf{\Lambda}}\bar{\mathbf{V}}^{\mathsf{T}}$, where $\mathbf{V}$ and $\bar{\mathbf{V}}$ contain in their columns the eigenvectors of $\mathbf{C}$ and $\bar{\mathbf{C}}$, respectively, and $\mathbf{\Lambda}$ and $\bar{\mathbf{\Lambda}}$ contain their eigenvalues on the diagonal. Following (Sihag et al., 2022, eq. (32)-(39)), under the assumption of small perturbation, i.e., $\|\mathbf{E}\| \ll 1$, the stability bound is the sum of three terms:

$$\mathbf{H}(\bar{\mathbf{C}})\mathbf{x} - \mathbf{H}(\mathbf{C})\mathbf{x} \approx \underbrace{\sum_{i=0}^{N}\tilde{x}_i\sum_{k=0}^{K}h_k\sum_{r=0}^{k-1}\mathbf{C}^r\lambda_i^{k-r-1}(\bar{\lambda}_i - \lambda_i)\mathbf{v}_i}_{\text{term 1}} \tag{10}$$

$$+ \underbrace{\sum_{i=0}^{N}\tilde{x}_i\sum_{k=0}^{K}h_k\sum_{r=0}^{k-1}\mathbf{C}^r\lambda_i^{k-r-1}(\lambda_i\mathbf{I}_N - \mathbf{C})(\bar{\mathbf{v}}_i - \mathbf{v}_i)}_{\text{term 2}} \tag{11}$$

$$+ \underbrace{\sum_{i=0}^{N}\tilde{x}_i\sum_{k=0}^{K}h_k\sum_{r=0}^{k-1}\mathbf{C}^r\lambda_i^{k-r-1}((\bar{\lambda}_i - \lambda_i)\mathbf{I}_N - \mathbf{E})(\bar{\mathbf{v}}_i - \mathbf{v}_i)}_{\text{term 3}} \tag{12}$$

where $\tilde{x}_i$ is the $i$-th component of the Covariance Fourier Transform of a generic graph signal $\mathbf{x}$, i.e., $\tilde{\mathbf{x}} = \mathbf{V}^{\mathsf{T}}\mathbf{x}$, and we inverted term 1 and 2 compared to (Sihag et al., 2022) for ease of explanation in the following. We now analyze the three terms separately.

**Term 1.** Leveraging (Sihag et al., 2022, eq. (59)-(61)), we have that

$$\sum_{i=0}^{N}\tilde{x}_i\sum_{k=0}^{K}h_k\sum_{r=0}^{k-1}\mathbf{C}^r\lambda_i^{k-r-1}(\bar{\lambda}_i - \lambda_i)\mathbf{v}_i = \sum_{i=0}^{N}\tilde{x}_i h'(\lambda_i)(\bar{\lambda}_i - \lambda_i)\mathbf{v}_i \tag{13}$$

where $h'(\lambda_i)$ is the derivative of the frequency response of the filter and is bounded in absolute value by $P$. Using Weyl's theorem (Golub & van Loan, 2013, Theorem 8.1.6), we have that $|\bar{\lambda}_i - \lambda_i| \leq \|\mathbf{E}\|$.

Therefore, by taking the norm and using the fact that $\sum_{i=1}^{N} |\tilde{x}_i| \leq \sqrt{N}\|\tilde{\mathbf{x}}\| = \sqrt{N}\|\mathbf{x}\| \leq \sqrt{N}$, we get

$$\left\|\sum_{i=0}^{N} \tilde{x}_i h'(\lambda_i)(\bar{\lambda}_i - \lambda_i)\mathbf{v}_i\right\| \leq \sum_{i=0}^{N} |\tilde{x}_i||h'(\lambda_i)|\|\mathbf{E}\|\|\mathbf{v}_i\| \leq \tag{14}$$

$$\sum_{i=0}^{N} |\tilde{x}_i|P\|\mathbf{E}\| \leq \tag{15}$$

$$P\sqrt{N}\|\mathbf{E}\|. \tag{16}$$

**Term 2**. Following (Sihag et al., 2022, eq. (40)-(54)), the norm of term 2 is bounded by

$$\left\|\sum_{i=0}^{N} \tilde{x}_i \sum_{k=0}^{K} h_k \sum_{r=0}^{k-1} \mathbf{C}^r \lambda_i^{k-r-1}(\lambda_i \mathbf{I}_N - \mathbf{C})(\bar{\mathbf{v}}_i - \mathbf{v}_i)\right\| \leq \sqrt{N} \sum_{i=1}^{N} |\tilde{x}_i| \max_j |h(\lambda_i) - h(\lambda_j)||\mathbf{v}_j^\mathsf{T}\bar{\mathbf{v}}_i| \tag{17}$$

where $h(\lambda)$ is the frequency response of the covariance filter. Note that in equation 17 we have a term $\sqrt{N}$ that does not appear in (Sihag et al., 2022, eq. (54)) because we consider operator norm instead of uniform norm. We now leverage the sin-theta theorem (Stewart & guang Sun, 1990, Theorem V.3.6): for $j = 1, \ldots, N$, we have that

$$\frac{1}{2}\|\mathbf{v}_j - \bar{\mathbf{v}}_j\|^2 = 1 - |\mathbf{v}_j^\mathsf{T}\bar{\mathbf{v}}_j| \leq \frac{\|\mathbf{E}\|^2}{\min_i |\lambda_i - \lambda_{i+1}|^2}. \tag{18}$$

By using Lemma 2, we obtain:

$$|\mathbf{v}_j^\mathsf{T}\bar{\mathbf{v}}_i| \leq \sqrt{2}\sqrt{1 - |\mathbf{v}_j^\mathsf{T}\bar{\mathbf{v}}_j|} \leq \frac{\sqrt{2}\|\mathbf{E}\|}{\min_i |\lambda_i - \lambda_{i+1}|} \quad \forall i \neq j. \tag{19}$$

By plugging equation 19 into equation 17, we have

$$\sqrt{N} \sum_{i=1}^{N} |\tilde{x}_i| \max_{j \neq i} |h(\lambda_i) - h(\lambda_j)||\mathbf{v}_j^\mathsf{T}\bar{\mathbf{v}}_i| \leq \tag{20}$$

$$\sqrt{2N}\|\mathbf{E}\| \sum_{i=1}^{N} |\tilde{x}_i| \max_{j \neq i} \frac{|h(\lambda_i) - h(\lambda_j)|}{\min_i |\lambda_i - \lambda_{i+1}|} \leq \tag{21}$$

$$\sqrt{2N}\|\mathbf{E}\| \sum_{i=1}^{N} |\tilde{x}_i| \max_{j \neq i} \frac{|h(\lambda_i) - h(\lambda_j)|}{|\lambda_i - \lambda_j|} \leq \tag{22}$$

$$\sqrt{2N}\|\mathbf{E}\| \sum_{i=1}^{N} |\tilde{x}_i|P \leq \tag{23}$$

$$\sqrt{2}N\|\mathbf{E}\|P \tag{24}$$

where we used the Lipschitz property of the filter and the fact that $\sum_{i=1}^{N} |\tilde{x}_i| \leq \sqrt{N}\|\mathbf{x}\|$.

**Term 3.** This term depends on the second-order error $((\bar{\lambda}_i - \lambda_i)\mathbf{I}_N - \mathbf{E})(\bar{\mathbf{v}}_i - \mathbf{v}_i)$. From Weyl's theorem (Golub & van Loan, 2013, Theorem 8.1.6), we have that $\|(\bar{\lambda}_i - \lambda_i)\mathbf{I}_N - \mathbf{E}\| \leq 2\|\mathbf{E}\|$. From equation 18, we have that $\|\bar{\mathbf{v}}_i - \mathbf{v}_i\| = \mathcal{O}(\|\mathbf{E}\|)$. Therefore, the product of the two is $\|(\bar{\lambda}_i - \lambda_i)\mathbf{I}_N - \mathbf{E}\|\|\bar{\mathbf{v}}_i - \mathbf{v}_i\| \approx \mathcal{O}(\|\mathbf{E}\|^2)$, which is dominated by the other two terms in equation 10 and equation 11 under small perturbation assumption.

We complete the proof by merging the terms in equation 24 and equation 16.

$\square$

## C.2 Proof of Thm. 2

From (Bickel & Levina, 2008b, Theorem 1), given a true covariance $\mathbf{C}$ and a hard-thresholded sample covariance $\bar{\mathbf{C}}$, it holds with probability $1 - o(1)$ that

$$\|\bar{\mathbf{C}} - \mathbf{C}\| = \|\mathbf{E}\| \le c_0 \sqrt{\frac{\log N}{t}}. \tag{25}$$

By replacing $\|\mathbf{E}\|$ in equation 9, the claim follows. $\qquad\square$

## C.3 Proof of Thm. 3

From (Deshpande & Montanari, 2016, Theorem 1), given a true covariance $\mathbf{C}$ and a soft-thresholded sample covariance $\bar{\mathbf{C}}$, it holds with probability $1 - o(1)$ that

$$\|\bar{\mathbf{C}} - \mathbf{C}\| = \|\mathbf{E}\| \le \frac{1}{\sqrt{t}} C c_0 \max(1, \lambda_0) \sqrt{\max\left(\log \frac{N}{c_0^2}, 1\right)} \tag{26}$$

for a generic constant $C$. By replacing $\|\mathbf{E}\|$ in equation 9, the claim follows. $\qquad\square$

## D Proof of Proposition 1

We have that

$$\left\|\mathbf{V}^\mathsf{T}\mathbf{x} - \bar{\mathbf{V}}^\mathsf{T}\mathbf{x}\right\| = \left\|\sum_{i=1}^N (\mathbf{v}_i - \bar{\mathbf{v}}_i)^\mathsf{T} x_i\right\| \le \sum_{i=1}^N \|\mathbf{v}_i - \bar{\mathbf{v}}_i\| |x_i|$$
$$\le \sum_{i=1}^N \|\mathbf{v}_i - \bar{\mathbf{v}}_i\| \le N \max_i \|\mathbf{v}_i - \bar{\mathbf{v}}_i\|. \tag{27}$$

where we used triangle inequality, the fact that $\|\mathbf{x}\| \le 1$ and the fact that $\|(\mathbf{v}_i - \bar{\mathbf{v}}_i)^\mathsf{T}\| = \|\mathbf{v}_i - \bar{\mathbf{v}}_i\|$. Using the sin-theta theorem (Stewart & guang Sun, 1990, Theorem V.3.6) (cf. equation 18), we have that $\max_i \|\mathbf{v}_i - \bar{\mathbf{v}}_i\| \le \sqrt{2}\|\mathbf{E}\|/\min_i |\lambda_i - \lambda_{i+1}|$ where $\|\mathbf{E}\| = \|\mathbf{C} - \bar{\mathbf{C}}\|$. Now, we use the expression for $\|\mathbf{E}\|$ with hard thresholding in (Bickel & Levina, 2008b, Theorem 1), i.e.,

$$\|\mathbf{E}\| \le c_0 \sqrt{\frac{\log N}{t}}, \tag{28}$$

and the claim follows. $\qquad\square$

## E Generalized Covariance Filter Frequency Response

We provide here additional details related to the derivation of the generalized covariance filter frequency response in Def. 6 following (Gao et al., 2021a). Consider a covariance filter operating on a sequence of $K$ sparsified covariances $\tilde{\mathbf{C}}_1 \ldots \tilde{\mathbf{C}}_K$ with eigendecomposition $\tilde{\mathbf{C}}_k = \tilde{\mathbf{V}}_k \tilde{\mathbf{\Lambda}}_k \tilde{\mathbf{V}}_k^\mathsf{T}$ (where $\tilde{\mathbf{V}}_k = [\tilde{\mathbf{v}}_{k1}, \ldots, \tilde{\mathbf{v}}_{kN}]$ and $\tilde{\mathbf{\Lambda}}_k = \mathrm{diag}(\tilde{\lambda}_{k1}, \ldots, \tilde{\lambda}_{kN})$) according to Def. 5. For matrix $\tilde{\mathbf{C}}_1$, we can express a signal $\mathbf{x}$ as $\mathbf{x} = \sum_{i_1=1}^N \hat{x}_{1i_1} \tilde{\mathbf{v}}_{1i_1}$, where $\hat{\mathbf{x}}_1 = [\hat{x}_{11}, \ldots, \hat{x}_{1N}]^\mathsf{T}$ is the covariance Fourier transform of $\mathbf{x}$ w.r.t. $\tilde{\mathbf{C}}_1$. By performing a graph signal shift, we obtain

$$\mathbf{x}^{(1)} = \tilde{\mathbf{C}}_1 \mathbf{x} = \tilde{\mathbf{C}}_1 \sum_{i_1=1}^N \hat{x}_{1i_1} \tilde{\mathbf{v}}_{1i_1} = \sum_{i_1=1}^N \hat{x}_{1i_1} \tilde{\lambda}_{1i_1} \tilde{\mathbf{v}}_{1i_1}. \tag{29}$$

When performing a second shift, i.e., $\mathbf{x}^{(2)} = \tilde{\mathbf{C}}_2 \tilde{\mathbf{C}}_1 \mathbf{x}$, we decompose each eigenvector $\tilde{\mathbf{v}}_{1i_1}$ by taking its covariance Fourier transform w.r.t. $\tilde{\mathbf{C}}_2$, i.e., we get $\tilde{\mathbf{v}}_{1i_1} = \sum_{i_2=1}^N \hat{x}_{2i_1i_2} \tilde{\mathbf{v}}_{2i_2}$ where

$\tilde{\mathbf{x}}_{2i_1} = [\hat{x}_{2i_11}, \ldots, \hat{x}_{2i_1N}]$ is the covariance Fourier transform of $\hat{\mathbf{v}}_{1i_1}$ over $\tilde{\mathbf{C}}_2$. Performing this operation on all eigenvectors $\tilde{\mathbf{v}}_{11}, \ldots, \tilde{\mathbf{v}}_{1N}$ and using equation 29, we can write

$$\mathbf{x}^{(2)} = \tilde{\mathbf{C}}_2 \sum_{i_1=1}^{N} \hat{x}_{1i_1} \tilde{\lambda}_{1i_1} \tilde{\mathbf{v}}_{1i_1} = \sum_{i_2=1}^{N} \sum_{i_1=1}^{N} \hat{x}_{2i_1i_2} \hat{x}_{1i_1} \tilde{\lambda}_{2i_2} \tilde{\lambda}_{1i_1} \tilde{\mathbf{v}}_{1i_1}. \tag{30}$$

Generalizing this to $k$ shifts, we get

$$\mathbf{x}^{(k)} = \sum_{i_k=1}^{N} \cdots \sum_{i_1=1}^{N} \hat{x}_{ki_{k-1}i_k} \ldots \hat{x}_{2i_1i_2} \hat{x}_{1i_1} \prod_{j=1}^{k} \tilde{\lambda}_{ji_j} \tilde{\mathbf{v}}_{ki_k} \tag{31}$$

and by aggregating the $K + 1$ shifted signals according to Def. 5 we obtain

$$\tilde{\mathbf{u}} = \sum_{i_K=1}^{N} \cdots \sum_{i_1=1}^{N} \hat{x}_{Ki_{K-1}i_K} \ldots \hat{x}_{2i_1i_2} \hat{x}_{1i_1} \sum_{k=1}^{K} h_k \prod_{j=1}^{k} \tilde{\lambda}_{ji_j} \tilde{\mathbf{v}}_{ki_k}. \tag{32}$$

From equation 32 we see that this representation involves all eigenvalues $\tilde{\mathbf{\Lambda}}_K, \ldots, \tilde{\mathbf{\Lambda}}_1$ and eigenvectors $\tilde{\mathbf{V}}_K, \ldots, \tilde{\mathbf{V}}_1$ of the sequence of covariance realizations. Therefore, we can consider the coefficients $\{\hat{x}_{1i_1}\}_{i_1=1}^{N}$ and $\{\hat{x}_{2i_ji_{j+1}}\}_{j=1}^{K-1}$ as the generalized covariance Fourier transform of signal $\mathbf{x}$ on the sequence of sparsified covariances $\tilde{\mathbf{V}}_K, \ldots, \tilde{\mathbf{V}}_1$. This supports the definition of a generalized frequency response for a covariance filter over random sparsified covariances in Def. 6.

# F  PROOF OF THM. 4

Consider a covariance filter $\mathbf{H}(\tilde{\mathbf{C}})\mathbf{x}$ operating on the matrix $\tilde{\mathbf{C}} = \mathbf{E} + \hat{\mathbf{C}}$, which represents a sparsified random matrix that is a copy of $\hat{\mathbf{C}}$ whose elements $c_{ij}$ are dropped with probability $(1 - p_{ij})$, where $\hat{\mathbf{C}}$ is the sample covariance matrix and $\mathbf{C}$ is the true covariance matrix. Then, the stability of the covariance filter depends on two sources of error. Indeed, by adding and subtracting $\mathbf{H}(\hat{\mathbf{C}})\mathbf{x}$ and leveraging the triangle inequality, we obtain

$$\|\mathbf{H}(\mathbf{C})\mathbf{x} - \mathbf{H}(\tilde{\mathbf{C}})\mathbf{x}\|^2 = \|\mathbf{H}(\mathbf{C})\mathbf{x} - \mathbf{H}(\hat{\mathbf{C}})\mathbf{x} + \mathbf{H}(\hat{\mathbf{C}})\mathbf{x} - \mathbf{H}(\tilde{\mathbf{C}})\mathbf{x}\|^2 \leq \tag{33}$$

$$\|\mathbf{H}(\mathbf{C})\mathbf{x} - \mathbf{H}(\hat{\mathbf{C}})\mathbf{x}\|^2 + \|\mathbf{H}(\hat{\mathbf{C}})\mathbf{x} - \mathbf{H}(\tilde{\mathbf{C}})\mathbf{x}\|^2 = \alpha + \beta \tag{34}$$

where $\alpha$ is an instability error due to the uncertainties in the covariance matrix estimate and $\beta$ depends on the stochastic sparsification of the sample covariance matrix. We analyze and provide an expression for these two terms in the remainder of the proof.

## F.1  COVARIANCE UNCERTAINTY ERROR

The effect of covariance estimation errors on the stability of VNN is deterministic and, consequently, can be bounded by the square of the bound in Thm. 1:

$$\|\mathbf{H}(\mathbf{C})\mathbf{x} - \mathbf{H}(\hat{\mathbf{C}})\mathbf{x}\|^2 \leq \beta^2 = \frac{P^2}{t} \mathcal{O}\left(N + \frac{\|\mathbf{C}\|^2 \log(Nt)}{\nu^2 t^2}\right) \tag{35}$$

where $t$ is the number of data samples used to estimate the covariance matrix.

## F.2  COVARIANCE SPARSIFICATION ERROR

To analyze the stability of VNN to the stochastic sparsification of the covariance matrix, we leverage and extend previous results on the stability of GNNs to stochastic graph perturbations. We begin by providing some lemmas that will be used in the main statement.

**Lemma 3.** *Consider a random covariance matrix $\tilde{\mathbf{C}}_r = \hat{\mathbf{C}} + \mathbf{E}_r$, where $\tilde{\mathbf{C}}_r$ is a copy of the sample covariance matrix $\hat{\mathbf{C}}$ whose elements $\hat{c}_{ij}$ are dropped independently with probability $1 - p_{ij}$ and $\mathbf{E}_r$ is its distance from $\hat{\mathbf{C}}$. Under the conditions of Def. 4, i.e., edge perturbations are undirected (i.e., $\mathbf{E}_r$ is symmetric) and no perturbations occur on the diagonal (i.e., $\mathbf{E}_r$ has zeros on the diagonal), it holds that*

$$\text{trace}(\mathbb{E}[\mathbf{E}_r^2]) = \sum_{i=1}^{N} \sum_{n=1}^{N} \hat{c}_{in}^2 (1 - p_{in}) = Q. \tag{36}$$

*Proof.* Each entry of the random error matrix $\mathbf{E}_r$ can be represented as $[\mathbf{E}_r]_{ij} = -\delta_{ij}c_{ij}$, where $\delta_{ij}$ is a Bernoulli variable that is one with probability $1 - p_{ij}$ and zero with probability $p_{ij}$. By performing the matrix multiplication, we can express each element of $\mathbf{E}_r^2$ and, consequently, its expectation, as

$$[\mathbf{E}_r^2]_{ij} = \sum_{n=1}^{N} \hat{c}_{in}\hat{c}_{nj}\delta_{in}\delta_{nj}, \quad [\mathbb{E}[\mathbf{E}_r^2]]_{ij} = \sum_{n=1}^{N} \hat{c}_{in}\hat{c}_{nj}\mathbb{E}[\delta_{in}\delta_{nj}]. \tag{37}$$

The Bernoulli variables $\delta_{ij}$ are independent except for $\delta_{ij} = \delta_{ji}$ given the symmetry of $\mathbf{E}_r$. Therefore, we have

$$\mathbb{E}[\delta_{in}\delta_{nj}] = \begin{cases} (1 - p_{in})(1 - p_{nj}) & \text{if } i \neq j \\ (1 - p_{in}) & \text{if } i = j \end{cases} \tag{38}$$

We can now compute the trace by summing the elements on the diagonal and using the fact that $\mathbf{E}_r$ is symmetric, i.e.,

$$\mathsf{trace}(\mathbb{E}[\mathbf{E}_r^2]) = \sum_{i=1}^{N}\sum_{n=1}^{N} \hat{c}_{in}\hat{c}_{ni}\mathbb{E}[\delta_{in}\delta_{ni}] = \hat{c}_{in}^2(1 - p_{in}). \tag{39}$$

$\square$

**Lemma 4.** *Consider a covariance filter $\mathbf{H}(\mathbf{C})$ with coefficients $\{h_k\}_{k=0}^{K}$ and generalized integral Lipschitz frequency response with constant $P$. Given some realizations of a random matrix $\tilde{\mathbf{C}}_r = \mathbf{E}_r + \hat{\mathbf{C}}$, for any signal $\mathbf{x}$, it holds that*

$$\mathbb{E}\left[\sum_{r=1}^{K}\mathsf{trace}\left(\sum_{k=r}^{K}\sum_{l=r}^{K} h_k h_l \mathbf{E}_r \hat{\mathbf{C}}^{k+l-2r} \mathbf{E}_r \hat{\mathbf{C}}^{r-1} \mathbf{x}\mathbf{x}^{\mathsf{T}} \hat{\mathbf{C}}^{r-1}\right)\right] \leq NP^2 \|\mathbf{x}\|^2 Q \tag{40}$$

*with $Q$ defined in equation 36.*

*Proof.* Following (Gao et al., 2021a, eq. (B.13)-(B.15)), we rewrite the term in equation 40 as

$$\sum_{i=1}^{N} \hat{x}_i^2 \sum_{r=1}^{K} \mathsf{trace}\left(\sum_{k=r}^{K}\sum_{l=r}^{K} h_k h_l \hat{\lambda}_i^{2r-2} \hat{\mathbf{C}}^{k+l-2r} \mathbb{E}[\mathbf{E}_r^2]\right) \tag{41}$$

where $\hat{x}_i$ is the $i$-th component of the Covariance Fourier Transform of the signal $\hat{\mathbf{x}} = \hat{\mathbf{V}}^{\mathsf{T}}\mathbf{x}$ and $\hat{\mathbf{C}} = \hat{\mathbf{V}}\hat{\mathbf{\Lambda}}\hat{\mathbf{V}}^{\mathsf{T}}$ is the eigendecomposition of $\hat{\mathbf{C}}$ with eigenvectors $\hat{\mathbf{V}} = [\hat{\mathbf{v}}_1, \ldots, \hat{\mathbf{v}}_N]^{\mathsf{T}}$ and eigenvalues $\hat{\mathbf{\Lambda}} = \mathsf{diag}(\hat{\lambda}_1, \ldots, \hat{\lambda}_N)$. Now, we leverage the property

$$\mathsf{trace}(\mathbf{A}\mathbf{B}) \leq \frac{\|\mathbf{A} + \mathbf{A}^{\mathsf{T}}\|}{2}\mathsf{trace}(\mathbf{B}) \leq \|\mathbf{A}\|\mathsf{trace}(\mathbf{B}) \tag{42}$$

which holds for any square matrix $\mathbf{A}$ and positive semi-definite matrix $\mathbf{B}$ (Wang et al., 1986). We note that $\mathbb{E}[\mathbf{E}_r^2]$ is positive semi-definite since $\mathbf{E}_r^2$ is the square of a symmetric matrix. Therefore, we use equation 42 to write

$$\sum_{i=1}^{N} \hat{x}_i^2 \sum_{r=1}^{K} \mathsf{trace}\left(\sum_{k=r}^{K}\sum_{l=r}^{K} h_k h_l \hat{\lambda}_i^{2r-2} \hat{\mathbf{C}}^{k+l-2r} \mathbb{E}[\mathbf{E}_r^2]\right) \leq \tag{43}$$

$$\sum_{i=1}^{N} \hat{x}_i^2 \left\|\sum_{r=1}^{K}\sum_{k=r}^{K}\sum_{l=r}^{K} h_k h_l \hat{\lambda}_i^{2r-2} \hat{\mathbf{C}}^{k+l-2r}\right\| \mathsf{trace}(\mathbb{E}[\mathbf{E}_r^2]) \tag{44}$$

where we also used the linearity of the trace operator to move the summation term.

From (Gao et al., 2021a, eq. (B.24)), we have that the term within the norm is bounded by the generalized Lipschitz coefficient of the filter, i.e.,

$$\left\|\sum_{r=1}^{K}\sum_{k=r}^{K}\sum_{l=r}^{K} h_k h_l \hat{\lambda}_i^{2r-2} \hat{\mathbf{C}}^{k+l-2r}\right\| \leq P^2. \tag{45}$$

Then, using Lemma 3, we have that

$$\text{trace}(\mathbb{E}[\mathbf{E}_r^2]) = \sum_{i=1}^{N} \sum_{n=1}^{N} \hat{c}_{in}^2 (1 - p_{in}) = Q. \tag{46}$$

Substituting these two terms in equation 44 leads to

$$\sum_{i=1}^{N} \hat{x}_i^2 P^2 Q \le N P^2 Q \sum_{i=1}^{N} \hat{x}_i^2 = N P^2 Q \|\mathbf{x}\|^2, \tag{47}$$

where we used the fact that $\sum_{i=1}^{N} \hat{x}_i^2 = \|\hat{\mathbf{x}}\|^2 = \|\mathbf{V}^\mathsf{T}\mathbf{x}\|^2 = \|\mathbf{x}\|^2$, i.e., the covariance Fourier transform does not modify the norm of the signal. $\qquad \square$

**Lemma 5.** *Consider two distinct random covariance realizations $\tilde{\mathbf{C}}_{r_1} = \hat{\mathbf{C}} + \mathbf{E}_{r_1}$ and $\tilde{\mathbf{C}}_{r_2} = \hat{\mathbf{C}} + \mathbf{E}_{r_2}$. Then, the following holds:*

$$\text{trace}(\mathbb{E}[\mathbf{E}_{r_1}\mathbf{E}_{r_2}]) = \sum_{i=1}^{N} \sum_{n=1}^{N} \hat{c}_{in}^2 (1 - p_{in})^2 \tag{48}$$

*Proof.* Similarly to Lemma 3, we can express each element of the expected value of the matrix product $\mathbf{E}_{r_1}\mathbf{E}_{r_2}$ as

$$[\mathbb{E}[\mathbf{E}_{r_1}\mathbf{E}_{r_2}]]_{ij} = \sum_{n=1}^{N} \hat{c}_{in}\hat{c}_{nj}\mathbb{E}[\delta_{r_1,in}\delta_{r_2,nj}] \tag{49}$$

where $\delta_{r,ij}$ is a Bernoulli variable relative to realization $\mathbf{E}_r$ that is one with probability $1 - p_{ij}$ and zero with probability $p_{ij}$. Since $\mathbf{E}_{r_1}$ and $\mathbf{E}_{r_2}$ are two different realizations, all variables $\delta_{r,ij}$ are independent. Therefore, $\mathbb{E}[\delta_{r_1,in}\delta_{r_2,nj}] = (1 - p_{in})(1 - p_{nj})$ and the trace becomes

$$\text{trace}(\mathbb{E}[\mathbf{E}_{r_1}\mathbf{E}_{r_2}]) = \sum_{i=1}^{N} \sum_{n=1}^{N} \hat{c}_{in}^2 (1 - p_{in})^2. \tag{50}$$

$\qquad \square$

**Lemma 6.** *Consider two distinct random covariance realizations $\tilde{\mathbf{C}}_{r_1} = \hat{\mathbf{C}} + \mathbf{E}_{r_1}$ and $\tilde{\mathbf{C}}_{r_2} = \hat{\mathbf{C}} + \mathbf{E}_{r_2}$. Then, the following holds:*

$$\text{trace}(\mathbb{E}[\mathbf{E}_{r_1}^2 \mathbf{E}_{r_2}]) = \mathcal{O}((1 - p_1)(1 - p_2)). \tag{51}$$

*where $p_1, p_2$ are two generic probabilities.*

*Proof.* From Lemma 3 we have that

$$[\mathbb{E}[\mathbf{E}_{r_1}^2]]_{ij} = \sum_{n=1}^{N} \hat{c}_{in}\hat{c}_{nj}\mathbb{E}[\delta_{r_1,in}\delta_{r_1,nj}]. \tag{52}$$

Therefore, the elements of $\mathbf{E}_{r_1}^2 \mathbf{E}_{r_2}$ are

$$[\mathbb{E}[\mathbf{E}_{r_1}^2 \mathbf{E}_{r_2}]]_{ij} = \sum_{m=1}^{N} \mathbb{E}[[\mathbf{E}_{r_1}^2]_{im}\hat{c}_{mj}\delta_{r_2,mj}] = \sum_{m=1}^{N} \sum_{n=1}^{N} \hat{c}_{in}\hat{c}_{nm}\hat{c}_{mj}\mathbb{E}[\delta_{r_1,in}\delta_{r_1,nm}\delta_{r_2,mj}]. \tag{53}$$

The three variables $\delta_{r_1,in}, \delta_{r_1,nm}, \delta_{r_2,mj}$ are all independent with the exception of $\delta_{r_1,in} = \delta_{r_1,ni}$ due to matrix symmetry. Therefore,

$$\mathbb{E}[\delta_{r_1,in}\delta_{r_1,nm}\delta_{r_2,mj}] = \begin{cases} (1 - p_{in})(1 - p_{nm})(1 - p_{mj}) & \text{if } i \neq m \\ (1 - p_{in})(1 - p_{mj}) & \text{if } i = m \end{cases} \tag{54}$$

We can now compute the trace:

$$\text{trace}(\mathbb{E}[\mathbf{E}_{r_1}^2\mathbf{E}_{r_2}]) = \sum_{i=1}^{N}\sum_{m=1}^{N}\sum_{n=1}^{N}\hat{c}_{in}\hat{c}_{nm}\hat{c}_{mi}\mathbb{E}[\delta_{r_1,in}\delta_{r_1,nm}\delta_{r_2,mi}]. \tag{55}$$

Therefore, the trace is a sum of terms of quadratic and cubic order in the probability value, i.e., $\mathcal{O}((1-p_1)(1-p_2))$ or $\mathcal{O}((1-p_1)(1-p_2)(1-p_3))$, where $p_1, p_2, p_3$ are generic probabilities values and the quadratic terms dominate the behavior since $1 - p < 1$. So,

$$\text{trace}(\mathbb{E}[\mathbf{E}_{r_1}^2\mathbf{E}_{r_2}]) = \mathcal{O}((1-p_1)(1-p_2)). \tag{56}$$

$\square$

**Lemma 7.** *Consider the sample covariance matrix $\hat{\mathbf{C}}$ and a random sparsified covariance $\tilde{\mathbf{C}}_r = \hat{\mathbf{C}} + \mathbf{E}_r$ as in Def. 4. We have that*

$$\text{trace}(\mathbb{E}[\mathbf{E}_r^3]) = \mathcal{O}((1-p_1)(1-p_2)). \tag{57}$$

*where $p_1, p_2$ are two generic probabilities.*

*Proof.* Similarly to Lemma 3, we can write the expected value of each element of $\mathbf{E}_r^3$ as

$$[\mathbb{E}[\mathbf{E}_r^3]]_{ij} = \sum_{m=1}^{N}\mathbb{E}[[\mathbf{E}_r^2]_{im}\hat{c}_{mj}\delta_{mj}] = \sum_{m=1}^{N}\sum_{n=1}^{N}\hat{c}_{in}\hat{c}_{nm}\hat{c}_{mj}\mathbb{E}[\delta_{in}\delta_{nm}\delta_{mj}] \tag{58}$$

and, consequently, the trace

$$\text{trace}(\mathbb{E}[\mathbf{E}_r^3]) = \sum_{i=1}^{N}\sum_{m=1}^{N}\sum_{n=1}^{N}\hat{c}_{in}\hat{c}_{nm}\hat{c}_{mi}\mathbb{E}[\delta_{in}\delta_{nm}\delta_{mi}] \tag{59}$$

where the Bernoulli variables $\delta_{in}, \delta_{nm}, \delta_{mi}$ are all independent with the exception of $\delta_{in} = \delta_{ni}$ due to matrix symmetry and the terms where $i = n = m = j$. Specifically, we have that

$$\mathbb{E}[\delta_{in}\delta_{nm}\delta_{mi}] = \begin{cases} (1-p_{in})(1-p_{nm})(1-p_{mi}) & \text{if } i \neq m \wedge i \neq n \\ (1-p_{in})(1-p_{mi}) & \text{if } i = m \oplus i = n \\ (1-p_{in}) & \text{if } i = m = n \end{cases} \tag{60}$$

where $\oplus$ denotes the xor operator. Now we note that, according to Def. 4, the matrix $\mathbf{E}_r$ has zeros on the diagonal or, equivalently, $1 - p_{ii} = 0 \quad \forall i$. As a consequence, the terms where $i = m = n$ in equation 59 are all zeros and the trace only contains terms of quadratic and cubic order in the probability value, i.e., $\mathcal{O}((1-p_1)(1-p_2))$ or $\mathcal{O}((1-p_1)(1-p_2)(1-p_3))$, where $p_1, p_2, p_3$ are generic probabilities values and the quadratic terms dominate the behavior since $1 - p < 1$. Therefore,

$$\text{trace}(\mathbb{E}[\mathbf{E}_r^3]) = \mathcal{O}((1-p_1)(1-p_2)). \tag{61}$$

$\square$

**Main statement.** Let $\mathbf{u} = \mathbf{H}(\hat{\mathbf{C}})\mathbf{x}$ and $\tilde{\mathbf{u}} = \mathbf{H}(\tilde{\mathbf{C}})\mathbf{x}$ be the outputs of the deterministic and stochastic filter, respectively. We are interested in the term

$$\mathbb{E}[\|\mathbf{u} - \tilde{\mathbf{u}}\|^2] = \mathbb{E}[\text{trace}(\mathbf{u}^\mathsf{T}\mathbf{u} + \tilde{\mathbf{u}}^\mathsf{T}\tilde{\mathbf{u}} - 2\mathbf{u}^\mathsf{T}\tilde{\mathbf{u}})] = \tag{62}$$

$$\mathbb{E}[\text{trace}(\tilde{\mathbf{u}}^\mathsf{T}\tilde{\mathbf{u}} - \mathbf{u}^\mathsf{T}\mathbf{u})] + 2\mathbb{E}[\text{trace}(\mathbf{u}^\mathsf{T}\mathbf{u} - \mathbf{u}^\mathsf{T}\tilde{\mathbf{u}})] \tag{63}$$

where we added and subtracted $\mathbf{u}^\mathsf{T}\mathbf{u}$ and used linearity of expectation and trace.

We represent a random covariance matrix as $\tilde{\mathbf{C}}_r = \hat{\mathbf{C}} + \mathbf{E}_r$, where $\mathbf{E}_r$ is a random matrix that contains the deviation from the true covariance. Following (Gao et al., 2021a, eq. (B.2)-(B.6)), we express the first term in equation 63 as

$$\mathbb{E}[\text{trace}(\tilde{\mathbf{u}}^\mathsf{T}\tilde{\mathbf{u}} - \mathbf{u}^\mathsf{T}\mathbf{u})] = -2\mathbb{E}[\text{trace}(\mathbf{u}^\mathsf{T}\mathbf{u} - \mathbf{u}^\mathsf{T}\tilde{\mathbf{u}})] + \tag{64}$$

$$\sum_{k=1}^{K}\sum_{l=1}^{K}h_k h_l \text{trace}\left(\mathbb{E}\left[\sum_{r=1}^{\min(k,l)}\hat{\mathbf{C}}^{k-r}\mathbf{E}_r\hat{\mathbf{C}}^{r-1}\mathbf{x}\mathbf{x}^\mathsf{T}\hat{\mathbf{C}}^{r-1}\mathbf{E}_r\hat{\mathbf{C}}^{l-r}\right]\right) + \tag{65}$$

$$\sum_{k=0}^{K}\sum_{l=0}^{K}h_k h_l \text{trace}(\mathbb{E}[\mathbf{S}_{kl}]) \tag{66}$$

where the term in equation 64 cancels out with the second term in equation 63, the term in equation 65 contains cross-products including two error matrices $\mathbf{E}_r$ with the same index $r$, and $\mathbf{S}_{kl}$ aggregates the sum of quadratic forms with two error matrices with different indices (i.e., terms of the form $f_1(\hat{\mathbf{C}})\mathbf{E}_{r_1}f_2(\hat{\mathbf{C}})\mathbf{E}_{r_1}f_3(\hat{\mathbf{C}})\mathbf{E}_{r_2}f_4(\hat{\mathbf{C}})\mathbf{E}_{r_2}f_5(\hat{\mathbf{C}})$ or $f_1(\hat{\mathbf{C}})\mathbf{E}_{r_1}f_2(\hat{\mathbf{C}})\mathbf{E}_{r_1}f_3(\hat{\mathbf{C}})\mathbf{E}_{r_2}f_4(\hat{\mathbf{C}})$ for two error matrices $\mathbf{E}_{r_1}$ and $\mathbf{E}_{r_2}$, $r_1 \neq r_2$). We now proceed to analyze the three terms in equation 64, equation 65, equation 66.

**First term.** The term in equation 64 is the opposite of the second term in equation 63, so it cancels out when substituted.

**Second term.** Analogously to (Gao et al., 2021a, eq. (B.8)), we rewrite the term in equation 65 leveraging the linearity of trace and expectation, the trace cyclic property $\mathsf{trace}(\mathbf{ABC}) = \mathsf{trace}(\mathbf{CAB}) = \mathsf{trace}(\mathbf{BCA})$ and rearranging the terms to change the sum limits. Then we exploit Lemma 4 to upper bound it as

$$\mathbb{E}\left[\sum_{r=1}^{K}\mathsf{trace}\left(\sum_{k=r}^{K}\sum_{l=r}^{K}h_k h_l \mathbf{E}_r \hat{\mathbf{C}}^{k+l-2r}\mathbf{E}_r\hat{\mathbf{C}}^{r-1}\mathbf{x}\mathbf{x}^{\top}\hat{\mathbf{C}}^{r-1}\right)\right] \leq NP^2\|\mathbf{x}\|_2^2 Q \qquad (67)$$

where $Q$ is defined in Lemma 4 and is a sum of terms linear in the probability value.

**Third term.** The term in equation 66 can be bounded by an expression similar to equation 44, but with at least two of the terms among $\{\mathbb{E}[\mathbf{E}_{r_1}^2], \mathbb{E}[\mathbf{E}_{r_2}^2], \mathbb{E}[\mathbf{E}_{r_1}], \mathbb{E}[\mathbf{E}_{r_2}]\}$ within the trace operator. Using Lemmas 5 to 7, we know that these trace terms are of the order $\mathcal{O}((1-p_1)(1-p_2))$ for two generic probability values $p_1, p_2$. Since the frequency response of the covariance filter $h(\boldsymbol{\lambda})$ is bounded, and consequently the coefficients $h_k$ are also bounded, we have that

$$\sum_{k=0}^{K}\sum_{l=0}^{K}h_k h_l \mathsf{trace}(\mathbb{E}[\mathbf{S}_{kl}]) = \mathcal{O}((1-p_1)(1-p_2)). \qquad (68)$$

By substituting the three bounds into equation 62, noticing that the terms of quadratic order $\mathcal{O}((1-p_1)(1-p_2))$ are dominated by the linear terms in $Q$ and using the fact that $\|\mathbf{x}\| \leq 1$, we obtain the final bound:

$$\mathbb{E}[\|\mathbf{u}-\tilde{\mathbf{u}}\|^2] \leq NP^2 Q + \mathcal{O}((1-p_1)(1-p_2)). \qquad (69)$$

$\square$

# G    SUFFICIENT CONDITION FOR PSD THRESHOLDED MATRIX

Consider a hard-thresholded sample covariance matrix $\bar{\mathbf{C}}$ as per Def. 2. Given the sample covariance matrix $\hat{\mathbf{C}}$, the following is a sufficient condition for $\bar{\mathbf{C}}$ to be positive semidefinite (PSD) (Bickel & Levina, 2008b, Section 2): $\bar{\mathbf{C}}$ is PSD if $\|\bar{\mathbf{C}} - \hat{\mathbf{C}}\| \leq \epsilon$ and $\lambda_{\min}(\hat{\mathbf{C}}) > \epsilon$ for an $\epsilon > 0$, where $\lambda_{\min}(\cdot)$ computes the smallest eigenvalue. That is, if the true covariance matrix is sparse, then the term $\|\bar{\mathbf{C}} - \hat{\mathbf{C}}\|$ decreases as the number of samples $t$ increases and the sparse estimate is more likely to be PSD.

# H    EXTENSION OF COVARIANCE SPARSIFICATION TECHNIQUES TO GRAPHS

The proposed sparsification techniques can also be applied to weighted graphs. Hard and soft thresholding of a graph consists of removing all the edges whose absolute weight is below a given threshold, and decreasing the weight of the others in case of soft thresholding. Stochastic sparsification can also be performed on graphs, where each edge $a_{ij}$ (where $a_{ij}$ is the absolute weight of the edge between nodes $i, j$ or 0 if the edge does not exist) is removed with a probability $1 - p_{ij}$. In the case of ACV, $p_{ij} = |a_{ij}| / \max_{i,j}|a_{ij}|$. For RCV, instead, we can control the level of sparsification through a parameter $p$ by defining an ordered set of probabilities $\mathcal{P} = \{p_1', \ldots, p_{N'}'\}$, where $p_i' \leq p_{i+1}'$, and each $p_i'$ is sampled from $\mathcal{N}(p, \sigma)$ with $\sigma = \min((1-p)/3, p/3)$ (such that the number of values not in $[0, 1]$, which we clip to the interval, is negligible) and $N'$ is the number of probability values for assignment. We then set $p_{ij} = p_k'$ where $k = |\{a_{lm} : |a_{lm}| < |a_{ij}|; l, m = 1, \ldots, N\}|$ is the position of $a_{ij}$ in the ordered ranking of absolute edge weights. This allows to increase the efficiency

Table 1: Details on datasets.

| Dataset | Nodes | Node features | Classes | Samples (train/valid/test) |
|---------|-------|---------------|---------|----------------------------|
| Epilepsy | 76 | 1 | 2 | 8000 (6400/800/800) |
| CNI | 200 | 122 | 2 | 240 (160/40/40) |
| MHEALTH | 24 | 128 | 12 | 5229 (3162/1003/1064) |
| Realdisp | 117 | 128 | 10 | 5831 (2651/1296/1884) |

of GNNs, which is a highly explored research direction (see, for example, Ye & Ji (2021); Peng et al. (2022); Rong et al. (2020)).

However, we here explicitly and purposely focus on VNNs and on covariance sparsification in order to, at the same time, improve VNN time efficiency and preserve the theoretical tractability of their stability with respect to covariance estimation errors, which improves with our techniques under the assumption of a sparse true covariance.

## I    EXPERIMENTAL SETUP AND ADDITIONAL RESULTS

### I.1    DATASETS

We provide additional details about the four real datasets we use in the experiments. Table 1 summarizes the main information about the datasets and Fig. 7 shows their empirical normalized covariance values distribution.

- **Epilepsy** (Kramer et al., 2008) consists of an electrocorticography time series collected during an epilepsy study at the University of California, San Francisco Epilepsy Center (Kramer et al., 2008). The dataset contains 8000 time series samples recorded at 76 locations in the brain, equally split before and after an epileptic seizure event. We use the same data preprocessing technique in (Natali et al., 2022), and we further z-normalize the samples of each class separately to prevent the classification task from being affected by trivial characteristics such as signal amplitude. We interpret each time series sample as a signal on a graph with 76 nodes, such that the covariance matrix represents brain functional connectivity. We perform binary classification to predict whether the sample was recorded before or after the seizure. We divide the datasets into train/validation/test splits of size 80%/10%/10%.

- **CNI** (Schirmer et al., 2021) was released for the Connectomics in NeuroImaging challenge and contains resting-state fMRI time series of patients with attention deficit hyperactivity disorder (ADHD) and neurotypical controls (NC). Each time series consists of 122 graph signals recorded in 200 brain locations (nodes) per patient. The available training set contains 200 patients and the test set 40 patients. We therefore consider a VNN with input feature size 122 over 200 nodes and we perform classification to predict whether the patient is ADHD or NC.

- **MHEALTH** (Banos et al., 2014a) contains measurements of wearable devices placed in the chest, right wrist and left ankle of 10 subjects performing 12 different actions. Since random data splits introduce contaminations due to overlapping windows (Tello et al., 2023), we use recordings from different subjects for train, validation and test. Specifically, we use subjects 6,10 for validation, 2,9 for test and the rest for training. We do not use the ECG measurements placed in the chest. The goal is to classify the action performed by the subject. The sampling rate is 50 Hz.

- **Realdisp** (Banos et al., 2014b) contains data of 17 subjects performing 33 actions. We use subjects 4, 6, 10 and 11 for validation; subjects 1, 7, 8, 9, 12 and 14 for test and the remaining for training. We use the 10 most common activity labels: walking, jogging, running, cycling, elliptic bike, trunk twist (arms outstretched), rowing, knees (alternatively) bend forward, waist bends forward, trunk twist (elbows bended). We segment the data in MHEALTH and Realdisp creating sliding windows of size 128 (2.56 seconds) for measurements relative to the same activity with 50% overlap.

Table 2: VNN hyperparameters on different datasets.

| Dataset | $L$ | $F_1\{\ldots, F_L\}$ | $K$ | Epochs | Learning rate | Batch size |
|---------|-----|----------------------|-----|--------|---------------|------------|
| Epilepsy | 5 | 32,32,32,32,32 | 1 | 200 | 0.001 | 100 |
| CNI | 1 | 64 | 1 | 50 | 0.015 | 50 |
| MHEALTH | 2 | 32,32 | 1 | 200 | 0.015 | 3162 |
| Realdisp | 2 | 128,64 | 1 | 500 | 0.001 | 5831 |
| SparseCov | 2 | 13,13 | 1 | 50 | 0.015 | 800 |
| LargeCov | 2 | 32,32 | 1 | 50 | 0.015 | 800 |
| SmallCov | 2 | 32,32 | 1 | 50 | 0.015 | 800 |

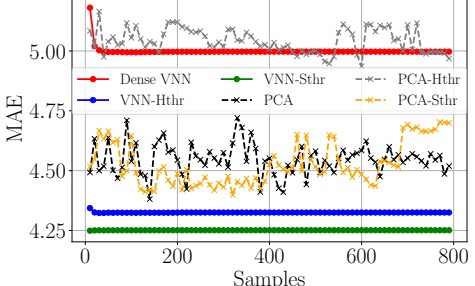 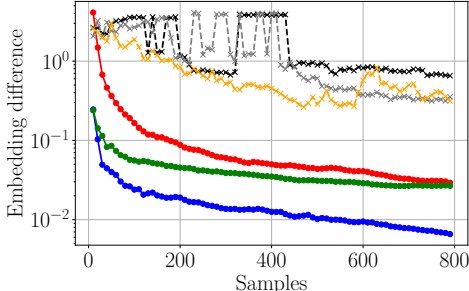

Figure 5: Stability of VNN and PCA-SVM with dense and sparse covariance on regression task for a synthetic dataset with sparse true covariance. (Left) Regression performance in terms of Mean Absolute Error. (Right) Embedding difference (i.e., $\|\Phi(\mathbf{x}, \bar{\mathbf{C}}, \mathcal{H}) - \Phi(\mathbf{x}, \mathbf{C}, \mathcal{H})\|$) between VNNs/PCA with true and estimated covariance. Standard deviations in our results are of order at most $10^{-2}$ for VNNs and $10^{-1}$ for PCA. Legend is shared.

### I.2 EXPERIMENTAL SETUP

**Hyperparameters.** Table 2 reports the hyperparameters of VNNs used for experiments on different datasets, which we find through a hyperparameter search. For regression and classification, we average the outputs of VNN over nodes and apply a 2-layer MLP for the final task. We use Adam optimizer with weight decay 0.001. For GraphSAGE in Table 3, we use the covariance matrix as a graph and we use the same hyperparameter configuration as VNN except for Realdisp, for which we use 2 layers of size 32 and 100 epochs.

**Implementation details.** We repeat all experiments 5 times and, for stochastic experiments, we sample 10 different sparsified covariance matrixes. We report average results and standard deviations. Experiments are run on a 13th Gen Intel Core i7-1365U CPU. We implement models using Pytorch and for GraphSAGE we use Pytorch Geometric (Fey & Lenssen, 2019). The sample covariance matrix $\hat{\mathbf{C}}$ is computed once over the complete training set and fixed during training, i.e., it is not affected by batching. For stability experiments, we use different sparsified covariances during test, whereas for the other experiments we keep the same matrix estimated on the training set.

### I.3 ADDITIONAL RESULTS

**Sparse true covariance.** Fig. 5 shows the embedding difference for VNNs and PCA variants with hard and soft thresholding. This confirms the observations in Sec. 5.1 relative to the increased stability of VNNs with thresholding w.r.t. PCA in sparse covariance settings since VNNs' outputs under sparsification are closer to the embeddings with the true covariance compared to PCA.

**Stochastic sparsification.** Fig. 6 shows the stability of VNNs under stochastic sparsification with different strategies on the 4 real-world datasets. On CNI, VNNs remain stable for all sparsification strategies: ACV achieves very close performance to dense VNN on both datasets and RCV performs closely for most values of $p$. This is consistent with the observations in Sec. 5.2, which suggest that most values in these covariances are not useful. On Epilepsy, MHEALTH and Realdisp, however, the sparsified VNNs appear less stable, despite most covariance values being small (cf. Fig. 7). This can

Table 3: Accuracy (%) on real datasets.

| | | Epilepsy | CNI | MHEALTH | Realdisp |
|---|---|---|---|---|---|
| **Baselines** | PCA + SVM | **100.0**±**0.0** | 50.0±0.0 | 17.7±0.0 | 11.9±0.0 |
| | Kernel PCA + SVM | **100.0**±**0.0** | 55.0±0.0 | 17.7±0.0 | 11.4±0.0 |
| | Sparse PCA + SVM | **100.0**±**0.0** | 55.0±0.0 | 17.7±0.0 | 11.8±0.0 |
| | GraphSAGE (Hamilton et al., 2018) | 57.9±0.7 | 55.0±0.7 | 77.0±1.9 | 67.3±1.2 |
| | Dense VNN (Sihag et al., 2022) | 98.6±0.4 | 54.0±3.8 | 87.6±3.6 | 71.6±4.0 |
| **Ours** | Hard-thr ($p = 0.75$) | 98.9±0.5 | 53.5±7.6 | 85.9±4.3 | 69.0±3.8 |
| | Hard-thr ($p = 0.5$) | 98.8±0.6 | 54.0±2.9 | 88.6±1.9 | 70.6±3.0 |
| | Hard-thr ($p = 0.25$) | 97.6±0.6 | 54.0±2.9 | 88.5±4.2 | 67.2±8.0 |
| | Soft-thr ($p = 0.75$) | 99.2±0.3 | 55.0±4.0 | 89.3±0.9 | 61.9±4.7 |
| | Soft-thr ($p = 0.5$) | 99.1±0.4 | 53.5±3.8 | 87.6±3.9 | 73.8±3.9 |
| | Soft-thr ($p = 0.25$) | 98.2±0.8 | 55.0±1.8 | 88.5±4.0 | 65.8±7.0 |
| | RCV ($p = 0.75$) | 98.0±1.2 | 56.0±6.5 | 90.4±5.3 | **72.3**±**2.8** |
| | RCV ($p = 0.5$) | 99.1±0.3 | 55.5±4.8 | 91.0±2.2 | 69.1±4.6 |
| | RCV ($p = 0.25$) | 99.4±0.3 | 55.0±1.8 | 88.1±3.1 | 71.6±1.0 |
| | ACV | 98.8±1.0 | **57.5**±**1.8** | **92.0**±**1.8** | 65.0±4.3 |

Table 4: Time (sec) for a batch forward pass on real datasets.

| | | Epilepsy | CNI | MHEALTH | Realdisp |
|---|---|---|---|---|---|
| **Baselines** | PCA + SVM | **0.35**±**0.02** | 0.03±0.02 | **0.08**±**0.01** | **0.14**±**0.01** |
| | Kernel PCA + SVM | 1.90±0.04 | 0.03±0.02 | 1.37±0.14 | 0.78±0.08 |
| | Sparse PCA + SVM | 0.57±0.07 | **0.02**±**0.00** | **0.08**±**0.00** | **0.14**±**0.01** |
| | GraphSAGE (Hamilton et al., 2018) | 4.47±0.18 | 0.65±0.18 | 0.70±0.08 | 60.6±9.7 |
| | Dense VNN (Sihag et al., 2022) | 4.49±0.01 | 0.61±0.08 | 1.58±0.07 | 72.6±1.6 |
| **Ours** | Hard-thr ($p = 0.75$) | 3.93±0.17 | 0.53±0.01 | 1.24±0.03 | 55.3±1.0 |
| | Hard-thr ($p = 0.5$) | 3.26±0.31 | 0.39±0.03 | 0.86±0.02 | 37.0±0.8 |
| | Hard-thr ($p = 0.25$) | 2.06±0.35 | 0.22±0.00 | 0.47±0.01 | 19.1±0.5 |
| | Soft-thr ($p = 0.75$) | 3.91±0.27 | 0.56±0.05 | 1.22±0.05 | 56.0±1.6 |
| | Soft-thr ($p = 0.5$) | 2.65±0.08 | 0.38±0.01 | 0.89±0.03 | 37.8±1.0 |
| | Soft-thr ($p = 0.25$) | 1.56±0.02 | 0.22±0.00 | 0.52±0.02 | 19.6±0.6 |
| | RCV ($p = 0.75$) | 3.90±0.15 | 0.54±0.01 | 1.29±0.06 | 60.1±0.5 |
| | RCV ($p = 0.5$) | 2.68±0.04 | 0.40±0.03 | 1.01±0.10 | 38.9±0.8 |
| | RCV ($p = 0.25$) | 1.73±0.01 | 0.23±0.01 | 0.64±0.04 | 21.6±1.0 |
| | ACV | 1.27±0.16 | 0.25±0.02 | 0.55±0.04 | 12.8±0.4 |

be due to the fact that the VNN on Epilepsy has 5 layers, which lowers stability according to Thm. 1, while the task for HAR datasets is more challenging compared to the task for brain datasets (in this case, there are 10 or 12 classes instead of the 2 for brain datasets), which may lead to lower model stability.

**Comparison with PCA and GraphSAGE on real datasets.** We report in Tables 3 and 4 the performance and time efficiency of S-VNNs, dense VNN, three different PCA-based classifiers (standard PCA, kernel PCA (Schölkopf et al., 1997) and PCA with hard-thresholded covariance matrix, all followed by a kernel SVM classifier) and GraphSAGE (Hamilton et al., 2018) operating on the covariance matrix as a graph as an example of sparsified GNN. For PCA-based classifiers, since PCA does not handle multiple features per node, for CNI, MHEALTH and Realdisp, we average the node features before classification. The results for S-VNNs and VNN are the same as Fig. 4.

PCA-based classifiers are fast, especially those with linear PCA, particularly on CNI, MHEALTH and Realdisp since they operate on the average of all node features. However, their performance is inconsistent. They perform comparably to S-VNNs on Epilepsy and CNI, but very poorly on the HAR datasets, as not considering multiple node features hinders their representation capabilities. This motivates the need for a more sophisticated and reliable approach such as VNNs. Furthermore, both sparse and non-linear PCA provably suffer from stability issues, reiterating the advantages of VNNs over them.

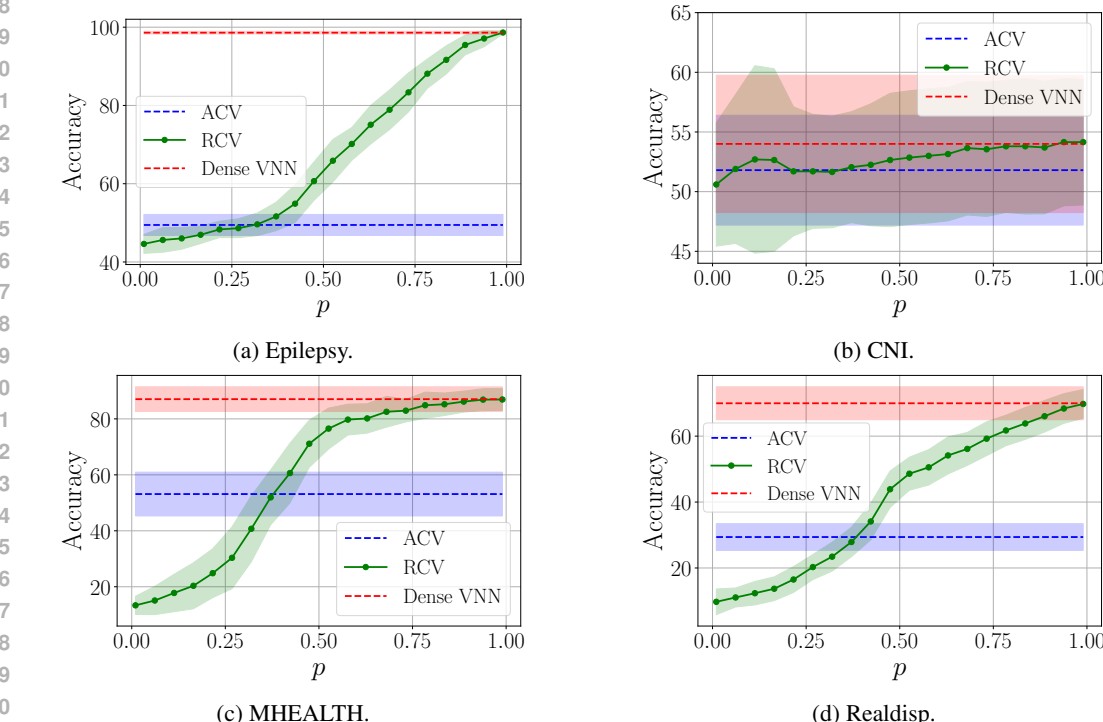

(a) Epilepsy.        (b) CNI.

(c) MHEALTH.        (d) Realdisp.

Figure 6: VNN stability (in terms of average accuracy and standard deviation) on real datasets for different stochastic covariance sparsification techniques.

GraphSAGE, instead, performs generally worse than S-VNNs (and especially poorly on Epilepsy) and it improves only slightly the time efficiency compared to dense VNN, whereas S-VNNs achieve a more consistent speed-up. This corroborates the effectiveness of our S-VNNs in both improving performance due to the removal of spurious correlation and increasing time efficiency.

**Remark 3.** *While the non-linear PCA and (sparse) VNNs may, at first sight, seem connected, they are fundamentally different since kernel PCA computes the principal directions of the data projected into a reproducing kernel Hilbert space, whereas VNNs manipulate the eigenvectors of the covariance matrix of the samples in the original space. Therefore, we merely use kernel PCA here as a baseline for additional comparison.*

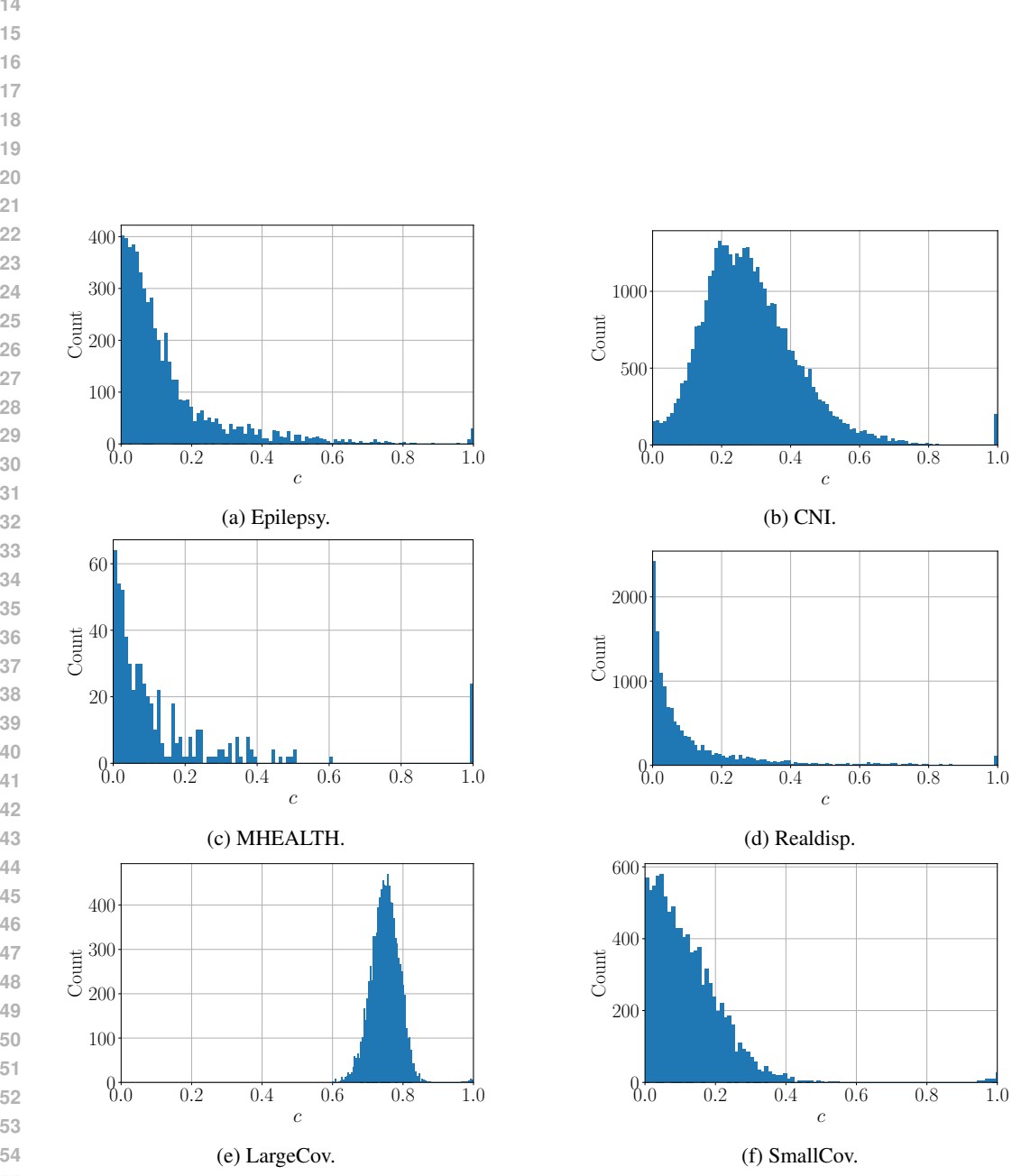

Figure 7: Covariance values distributions for real and synthetic datasets.

