# OpenReview forum: "Sparse Covariance Neural Networks"
_ICLR.cc/2025/Conference — Submitted to ICLR 2025_

### Official Review · Reviewer_rRYr · 2024-11-03

**Soundness:** 3
**Presentation:** 3
**Contribution:** 2
**Rating:** 5
**Confidence:** 4

**Summary:**

This paper studies the sparsification strategy upon the spurious correlation and computation efficiency issues for the covariance matrix used in the coVariance Neural Networks (VNN), which is more stable than the Principal component analysis (PCA) methods in the previous studies. This paper proposes hard and soft thresholding strategies if the true covariance matrix is sparse, and two stochastic sparsification techniques, including Absulute covariance values (ACV) and Ranked covariance values (RCV) when the covariance matrix is dense, and theoretically analyze the sparsification error and covariance uncertainty for the stability of VNN. The effectiveness of sparsity techniques is validated through synthetic and real datasets in contrast to the dense-VNN and sparse-PCA on the performance and computational time.

**Strengths:**

Quality: The paper is well-written, the motivation sounds reasonable, and the proposed sparsification strategies seem good with a theoretical analysis of the stability.
Originality: The paper proposes several sparsification strategies to tackle the spurious correlation and computation cost issues for the VNN. Although the theoretical analysis of the stability of VNN is good, the originality of the proposed solutions is limited.
Significance: Spurious correlation and sparsity techniques are important.

**Weaknesses:**

The novelty of the proposed strategies is limited, similar strategies have been used in the study of neural networks like dropout or pruning, and the current results are not enough strong.

**Questions:**

Although current results can validate the effectiveness of the proposed strategies, the results should be compared with the covariance with sparsity regularizer, not just dense-VNN and robust-PCA.

---

> ### Author Response · Authors · 2024-11-20
>
> We thank the reviewer for their analysis. We respond to the comments in the following.
>
> **The novelty of the proposed strategies is limited, similar strategies have been used in the study of neural networks like dropout or pruning, and the current results are not enough strong.**
>
> We refer the reviewer to the general comment of the rebuttal, where we reiterate the relevance of our contributions. Mostly, our contributions do not lie only in the proposed sparsification strategies, but also in their theoretical connection with VNN stability, which is new and unexplored.
>
> Furthermore, our approach is different from dropout and pruning techniques, since the latter remove neural network weights to increase efficiency or prevent overfitting. We, instead, consider the sparsification of the covariance matrix, but the number of neural network weights stays the same. There exist graph sparsification approaches akin to dropout (e.g., [3,4]), which we discuss briefly in our Related Works, but they mainly focus on oversmoothing or expressivity, meanwhile our focus is on stability and computation efficiency.
>
>
> **Although current results can validate the effectiveness of the proposed strategies, the results should be compared with the covariance with sparsity regularizer, not just dense-VNN and robust-PCA.**
>
> We are not sure what the reviewer refers to with "covariance with sparsity regularizer", since the thresholding techniques in Def. 2,3 can also be interpreted as sparsity regularizers.
> If the reviewer refers to covariance estimation strategies that enforce sparsity in the objective function of the covariance estimation problem (e.g., [1,2]), we opted for not including them as they might incur in the significant overhead of solving an optimization problem, thus defeating the purpose of increased time and computation efficiency. Furthermore, it might be harder to theoretically characterize the impact of these sparsity strategies on the VNN behavior.
>
>
> **Conclusion**
>
> We hope to have made the relevance of our contributions more clear, and we remain available for further discussion.
>
>
>
> **References**
>
> [1] Bien et al. Sparse Estimation of a Covariance Matrix, Biometrika 2011.
>
> [2] Friedman et al., "Sparse inverse covariance estimation with the graphical lasso", Biostatistics, 2008.
>
> [3] Pál András Papp, Karolis Martinkus, Lukas Faber, and Roger Wattenhofer. Dropgnn: Random dropouts increase the expressiveness of graph neural networks. Advances in Neural Information Processing Systems, 2021.
>
> [4] Yu Rong, Wenbing Huang, Tingyang Xu, and Junzhou Huang. Dropedge: Towards deep graph convolutional networks on node classification. In International Conference on Learning Representations, 2020.

---

> > ### Comment · Reviewer_rRYr · 2024-11-26
> >
> > Apologies for my late response, and thank the author for answering my main concerns.
> > For the first point, my previous comments already mentioned the theoretical analysis of the stability of VNN. I still hold it as incremental work because the acceleration mainly comes from the more zero terms in the covariance matrix. The proposed strategy to obtain the sparse matrix is random methods, which can also be viewed as a perturbation and that's why stability analysis is needed. And the current results are not strong enough.
> > For the second point, sorry for my unclarity on "covariance with sparsity regularizer", here means to obtain a covariance matrix with L1 constraints from the sample data. Since the paper studies the random sparsity techniques on VNN, I think it would be more persuasive to compare the results with the VNN model using the sparse covariance matrix.
> >
> > After reading the author's responses and the other reviews, I have decided to keep my score.

---

### Official Review · Reviewer_KsmG · 2024-11-03

**Soundness:** 2
**Presentation:** 3
**Contribution:** 2
**Rating:** 3
**Confidence:** 4

**Summary:**

This paper presents a sparse version of covariance neural networks (VNNs), showing that the sparse-VNNs are more stable than nomial VNNs and sparse component analysis.

**Strengths:**

The presentation of the paper is clear.

**Weaknesses:**

1. It is well known that the covariance matrix is not invariant to the scale of the data, making it impractical to set a common threshold for different elements of c_{ij}. Thus, the rationale behind Definition 2 is difficult to justify.

2. From a sparsity perspective, the elements c_{ij}'s should be classified into two categories: zero and nonzero. However, this key point is obscured in Theorem 2, making the results difficult to interpret. In other words, it is challenging to justify that the proposed method effectively denoises the data.

3. The contribution of the paper appears incremental in light of the existing work by Sihag et al. (2022).

**Questions:**

1. See weakness.

2. Additionally, using t to represent the sample size is somewhat unusual to me, though this is merely a notation issue.

---

> ### Author Response · Authors · 2024-11-20
>
> We thank the reviewer for analyzing our paper. We address the comments in the following.
>
> **It is well known that the covariance matrix is not invariant to the scale of the data, making it impractical to set a common threshold for different elements of $c_{ij}$. Thus, the rationale behind Definition 2 is difficult to justify.**
>
> While this is true for unscaled data, in practice it is very common to rescale data to feature-wise unit variance before applying machine learning techniques, which allows to set a common threshold for different elements. The hard and soft thresholding in Def. 2,3 are also common in related works [1,2].
>
> **From a sparsity perspective, the elements $c_{ij}$'s should be classified into two categories: zero and nonzero. However, this key point is obscured in Theorem 2, making the results difficult to interpret. In other words, it is challenging to justify that the proposed method effectively denoises the data.**
>
> The definition of the sparse covariance class $\mathcal{C}$ in Thm. 2 considers matrixes with at most $c_0$ non-zero elements in each row. Sparsifying the estimate through thresholding reduces the number of non-zero elements according to the threshold, which leads to better estimates as shown in [1,2].
>
> **The contribution of the paper appears incremental in light of the existing work by Sihag et al. (2022).**
>
> We refer the reviewer to our general comment, where we reiterate why we believe our contribution to be more than incremental.
>
> **Conclusion**
>
> We believe to have effectively addressed the concerns of the reviewer about the weaknesses 1 and 2, and we hope that the relevance of our contributions are more clear through our general comment. We remain available for further discussion.
>
> **References**
>
> [1] Peter J Bickel and Elizaveta Levina. Covariance regularization by thresholding. The Annals of Statistics, 2008.
>
> [2] Yash Deshpande and Andrea Montanari. Sparse pca via covariance thresholding. Journal of Machine Learning Research, 2016.

---

### Official Review · Reviewer_9p73 · 2024-11-05

**Soundness:** 2
**Presentation:** 2
**Contribution:** 2
**Rating:** 3
**Confidence:** 3

**Summary:**

**Summary:**
Graph convolutions on the covariance matrix of tabular data are performed by a architecture called Covariance Neural Networks (VNNs). These rely an empirical estimates of the covariance matrix, which is notoriously difficult. The authors propose Sparse VNNs, which sparsifies the covariance matrix before convolution is applied. This is advantageous when the true covariance is sparse, and also when the true covariance is dense, and different techniques are investigated. S-VNNs are compared against VNNs and PCA. Experiments are provided on brain data and human action reconition, showing performance, sensitivity and efficiency benefits over VNNs.

**Strengths:**

**Strengths:**
- The proposed architectures come with apparent theoretical guarantees on the closeness of hidden representations or predictions when applied to the approximate versus true covariance matrices. These distances become smaller as the number of samples increase, at a rate of $\mathcal{O}(t^{-1/2})$, ignoring some parameter-dpendent constants.
- Experiments are provided showing the predictive performance and stability of the proposed methods. This experiments agree with the theory, and nicely demonstrate the approach.
- A very large amount of related literature is cited, and this seems appropriate.
- The theory appears to be mostly sound, roughly based on a Lipschitz assumption. (although there are some isolated places where precision could be improved, see weaknesses below).

**Weaknesses:**

**Weaknesses:**
- Comparing Lemma 1 and Thoerem 1. I am guessing (please correct me if I am wrong!) that somehow hidden inside $\mathcal{O}$ in Theorem 1 are the parameters $\mathcal{H}$. Intuitively, these should learn something similar to PCA, and if the eigenvalues of PCA are close, this constant term in $\mathcal{O}$ will be bad but in terms of $\mathcal{H}$. In Lemma 1, the constant term in terms of the small gap eigenvalues is explicitly given, and it is obvious how this causes instability. Whether my guess about Theorem 1 is correct or not, the authors should state either way about the possibility of dependent of $\mathcal{H}$ on the factors in $\mathcal{O}$. Right now the claim that " VNNs do not suffer from this as the covariance filter can exhibit a stable response to close eigenvalues at the expense of lower discriminability" is not clear, as are similar repeated claims throughout the paper. It seems as though this is also an important consideration for later derived theoretical results (Theorem 2, Proposition 1, ...). **This is my most major important concern.**
- Theorem 1 explicitly writes the probability of the event. The later results only state with high probability. What does with high probability mean?
- In definition 3, how does one ensure that the resulting soft-thresholded matrix is PSD?
- In definition 4, how does one ensure that the matrix is PSD after dropout?
- What happens when the true data distribution is heavy tailed, and does not have a (finite) covariance matrix? I guess there should be some condition in the theoretical results, which is currently absent, on the distribution of $x$. In Theorem 1 a Gaussian distribution is used (which has finite variance), and it is not clear in the presentation of later results if a Gausssian is also being used.
- It is claimed that "For sparse covariance, ... the hard thresholding improves stability." According to my understanding, the authors are able to derive bounds which are tighter for sparse covariances than for dense covariances with hard thresholding. This does not necessarily show that hard thresholding definitively improves stability, only the bound is tighter. If I have understood correctly, perhaps the authors should rephrase their claim (otherwise, am happy to be corrected).



**Minor:**
- I don't understand one sentence in Theorem 1. "Consider a generic data sample $x \sim \mathcal{N} (0, C)$ such that $\Vert x \Vert \leq 1$". Does this mean $x$ is drawn from the conditional distribution obtained by conditioning a Gaussian random vector on the event that the norm is less than or equal to 1? I believe this should be phrased better, in terms of the distribution x is actually drawn from. (I understand this is from another paper, but still it would be nice to improve clarity here).
- Incorrect grammar for paragraph starting line 167/168.
- Limitations of PCA in terms of unstable or poor estimation of eigenvalues is discussed. Do the authors know where does this fit into more robust variants of PCA, exponential family PCA, kernel PCA, etc.? Do such advanced variants of PCA overcome these issues? Either way, it would be nice to mention in the discussion in a sentence around line 39/40.

**Questions:**

Please address each of the weaknesses above. In particular:
- In Theorem 1 and related results, is it true that $\mathcal{O}$ hides dependence on $\mathcal{H}$, which as in the case of PCA, could be arbitrarily bad?
- What does with high probability mean exactly?
- How are matrices ensured to be PSD?
- What happens in the case of undefined / infinite variance? How should this be reflected in the theorems? Is a Gaussian assumption on x used throughout, or only in the first result?
- Does the theoretical result really show a guaranteed improvement in stability? Or does it only fail to show a decrease in stability?
- Please clarify whether $x$ is drawn from an (unconditional) Gaussian or from a "truncated" Gaussian conditioned on its norm.

---

> ### Author Response · Authors · 2024-11-20
>
> We thank the reviewer for the thorough comments provided on our work. We address them in the following.
>
> **Where are the parameters $\mathcal{H}$ in the stability bound?**
>
> The parameters $\mathcal{H}$ are filter coefficients that determine the frequency response of the covariance filter (lines 167-171). Shortly, the operation of the covariance filter can be modeled by a polynomial of the covariance eigenvalues $\lambda$ in the spectral domain, where $\mathcal{H}$ are the polynomial coefficients. The variability of the filter frequency response is bounded by the Lipschitz constant $P$ as per Def. 1. Therefore, the filter coefficients $\mathcal{H}$ as well as the eigenvalue gap $\min_i|\lambda_i-\lambda_{i+1}|$  are "absorbed" by the Lipschitz constant $P$ in Thm. 1 and they are **not** in the term within $\mathcal{O}$.
>
> This leads to a similar expressivity vs stability tradeoff as that in GNNs [1]. A larger $P$ allows for a more expressive polynomial with larger variability for close eigenvalues, but corresponds to more different responses when the input is perturbed, while a smaller $P$ guarantees close responses in case of perturbations but lowers the model expressivity.
>
> **Theorem 1 explicitly writes the probability of the event. The later results only state with high probability. What does with high probability mean?**
>
> We thank the reviewer for pointing this out. The bounds in Thm. 2 and 3 hold stochastically, i.e., with a probability $1 - o(1)$ where $o(1)$ gets smaller as the number of data samples $t$ increases. We do not provide an explicit expression for the term $o(1)$ due to its complexity, as done also in [2,3] which we base our stability analysis upon. We refer to the proofs in [2,3] for additional details on the probability terms. We have clarified this in the revised manuscript.
>
> **In definition 3, how does one ensure that the resulting soft-thresholded matrix is PSD?** **In definition 4, how does one ensure that the matrix is PSD after dropout?**
>
> As we elaborate in Remark 2, both thresholding and covariance sparsification do not ensure that the resulting matrix is PSD. This is not needed for our theoretical results and is common in related works (e.g., [2,3]). We also report in Appendix G a sufficient condition for a hard-thresholded covariance matrix to be PSD.
>
> **What happens when the true data distribution is heavy tailed**
>
> The theoretical results hold for any random variable $\mathbf{x}$, and the Gaussian assumption is Thm. 1 is actually not needed (we have removed it from the revised manuscript). In particular, we always compute the covariance matrix $\mathbf{C}$ for the random variable $\mathbf{x}$ regardless of its distribution. We are not sure what the reviewer refers to with a random variable not having a finite covariance matrix, and we are happy to discuss this further if the reviewer can point us to a reference to understand this concept.
>
> **Wrong claim about increased stability**
>
> The reviewer is right that our analysis under hard thresholding leads to a smaller bound, but not definitely guarantees better stability. We have rephrased this in the revised manuscript.
>
> **Distribution of $\mathbf{x}$ in Thm. 1**
>
> $\mathbf{x}$ does not need to follow a Gaussian distribution but only needs to have covariance $\mathbf{C}$. The assumption on its norm, $\\|\mathbf{x}\\| \leq 1$, is only used to make the bound more concise and we can derive a similar bound by replacing $1$ with some constant $C_\text{norm}$ without loss of generality. Furthermore, we can always rescale data to have a unit bound on the norm. We have clarified this in the revised manuscript.
>
> **Incorrect grammar for paragraph starting line 167/168.**
>
> We have rephrased this sentence as follows: "We now define the frequency response of a covariance filter, which will be instrumental for our analysis."
>
> **Do advanced PCA variants overcome instability issues w.r.t. sample estimation?**
>
> The instability issues related to sample estimation of PCA are present in kernel PCA as well. Indeed, kernel PCA also estimates a sample covariance matrix after mapping the samples to a kernel space, and this estimation is prone to errors.
> Other extensions of PCA, such as exponential PCA, on the other hand, might be less prone to estimation errors, but their convergence properties as the number of samples increases are more difficult to characterize. Nevertheless, standard PCA remains highly popular in practice, which justifies it being the main focus of our analysis.
>
>
> **Conclusion**
>
> We thank the reviewer for pointing out unclear passages in our work. We have revised the manuscript accordingly and we believe that now clarity is improved, and we hope to have effectively addressed the reviewer's concerns. We remain available for further discussions and questions.

---

> > ### Author Response · Authors · 2024-11-20
> >
> > **References**
> >
> > [1] Fernando Gama, Joan Bruna, and Alejandro Ribeiro. Stability properties of graph neural networks. IEEE Transactions on Signal Processing, 2020.
> >
> > [2] Peter J Bickel and Elizaveta Levina. Covariance regularization by thresholding. The Annals of Statistics, 2008.
> >
> > [3] Yash Deshpande and Andrea Montanari. Sparse pca via covariance thresholding. Journal of Machine Learning Research, 2016.

---

> > ### Comment · Reviewer_9p73 · 2024-11-26
> >
> > Apologies for my late response. Thanks in particular for addressing my most major concern.
> >
> > To respond to "We are not sure what the reviewer refers to with a random variable not having a finite covariance matrix, and we are happy to discuss this further if the reviewer can point us to a reference to understand this concept." Some random variables can have an undefined or infinite covariance matrix. Such random variables appear for example in the prior/posterior/empirical distribution of neural nets trained in a "non-lazy" regime. They also appear in many data generating processes in nature.
> >
> > After reading the author responses and the other reviews, I have decided to keep my score.

---

### Official Review · Reviewer_G6qo · 2024-11-05

**Soundness:** 2
**Presentation:** 1
**Contribution:** 1
**Rating:** 1
**Confidence:** 3

**Summary:**

This paper builds upon covariance neural networks, which are constructed to process covariance matrices. The authors study the impact of sparsifying the covariance matrix with hard or soft thresholding before feeding it to the network. They demonstrate that this improves the estimation when the true covariance matrix is sparse, and explain how to drop coefficients at random when it is not the case. The authors demonstrate some improvements compared to no thresholding in several experiments.

**Strengths:**

- The idea of sparsifying the covariance before feeding it to the network is sound.

**Weaknesses:**

- This paper is very hard to follow. Several variables are not defined ($V$, $u$, $u_g$, $h_{klfg}$, etc...). Several concepts must be guessed from the text (what are the covariance filters? how does $u$ relate to the $u_g$ in eq. 1? what are the per covariance filters in theorem 1?). Overall, it is hard to understand this paper alone without reading the original paper on covariance neural networks. This paper needs a major rewriting before being ready for publication, which explains my note.
- The novelty is minor. This paper builds upon covariance networks, which are seldom used in practice, and incrementally improves on them by pre-processing the covariances.

**Questions:**

- How can we know in practice if the covariance of the dataset is sparse?
- How does $\nu$ in thm.1 depends on the data distribution? I only see one data distribution, which depends only on $C$, so $\nu$ is only a function of $C$?
- In def. 1, is it for a fixed matrix? In which set do the eigenvalue pairs $\lambda_i, \lambda_j$ belong? This is unclear.
- Is lemma 1 an original result? If not then a citation is needed here.
- Should $F_{in} = F_{out}$ in (1)? since it seems like $u^l$ is both of size $F_{out}$ and $F_{in}$.

---

> ### Author Response · Authors · 2024-11-20
>
> We wish to thank the reviewer for the comments on our paper. We regret to see that our contributions have not been appreciated, and we hope to make them more explicit and effectively address the reviewer's concerns in the following.
>
> ### Weakness 1: the paper is hard to follow
>
> We believe this weakness is due to some notation conventions that are common in some related works [1,2,3], but might lead to confusion. We have taken care of clarifying the meaning of each variable in the revised manuscript. More precisely:
> - Several variables are not defined ($\mathbf{V}, \mathbf{u}, \mathbf{u_g}, h_{klfg}$, ... ).
> We have defined all these variables in the revised manuscript. In particular, $\mathbf{V}=[\mathbf{v}\_0,...,\mathbf{v}\_{N-1}]$ contains the eigenvectors of the true covariance matrix. We modified eq. (1) to clarify that $\\{\mathbf{u}^{l}\_f\in\mathbb{R}^{N}\\}\_{f=1}^{F\_l}$ are the outputs of the $l$-th VNN layer, each produced by the $f$-th covariance filter bank. This filter bank is composed of $F_{l-1}$ covariance filters, each processing the signals at the previous layer $\\{\mathbf{u}\_g^{l-1}\in\mathbb{R}^{N}\\}\_{g=1}^{F\_{l-1}}$ separately.
>  At the first layer, we have $\\{\mathbf{u}\_g^0 = \mathbf{x}\_g\\}\_{g=1}^{F\_0}$ where $F_0$ is the node feature size. $h_{klfg}$ are the network learnable parameters for the $k$th order, $l$th layer, $g$th input signal and $f$th filter output in eq. (1).
>
> - What are the covariance filters?
> We have defined covariance filters in line 88 of the original manuscript (line 86 in the revised one).
> - How does $\mathbf{u}$ relate to $\mathbf{u_g}$ in eq. 1?
> We have clarified this in the answer above.
> - What are the per covariance filters in theorem 1?
> The covariance filters in Theorem 1 are those defined in eq. (1) at each layer, which compose a VNN.
>
> ### Weakness 2: minor novelty
>
> We refer the reviewer to the general comment of this rebuttal, in which we reiterate our contributions and novelty. Specifically, in the section "Significance of covariance neural networks" we point out why VNNs are not "seldom used in practice", as they represent a general framework which encompasses several methods that commonly applied in practice.
>
> ### Questions
>
> - How can we know in practice if the covariance of the dataset is sparse?
> The sparsity of the covariance matrix is a common and reasonable assumption in a variety of applications. For example, in brain signal processing, it is known that only some areas of the brain activate simultaneously for different tasks, which leads to sparse correlations [7]. In financial data, only some stocks are affected by similar factors and therefore present analogous variations, which again leads to sparse correlations [8]. In all these cases, the sample covariance matrix might still be dense due to spurious correlations, which hinder the quality of the VNN performance and motivate the proposed S-VNN. Furthermore, even in other cases in which it is difficult to assume covariance sparsity a priori, the improved performance of sparsity-based regularizers leads to a belief that it is a good assumption [4,5]. We have clarified this aspect in the revised manuscript.
>
> - How does $\nu$ in thm.1 depends on the data distribution? I only see one data distribution, which depends only on $\mathbf{C}$, so $\nu$ is only a function of $\mathbf{C}$?
> We thank the reviewer for pointing out this inconsistency. The assumption that $\mathbf{x}$ is Gaussian is actually not needed for the proof and we have removed it from the revised manuscript. The relation between $\nu$ and the data distribution is analyzed in [1] and we did not report details in our paper as it is not the focus of our analysis. Briefly, $\nu$ relates to the kurtosis of the data distribution, which in turns relates to the tail of the covariance eigenvalues distribution.
>
> - In def. 1, is it for a fixed matrix? In which set do the eigenvalue pairs $\lambda_i,\lambda_j$ belong? This is unclear.
> We have clarified in the revised manuscript that the Lipschitz condition is for any $\lambda_i, \lambda_j \in [0,\lambda_\textnormal{max}], \lambda_i \neq \lambda_j$, where $\lambda_\textnormal{max}$ identifies a suitable range in which the covariance eigenvalues lie and can be application-specific (e.g., $\lambda_\textnormal{max}=1$ in case the covariances are normalized by their largest eigenvalue).
>
>
> - Is lemma 1 an original result? If not then a citation is needed here.
> Lemma 1 is an original result derived based on the results in [1] and [6], so we did not report a citation for it.
>
> - Should $F_{in}=F_{out}$ in (1)? since it seems like $u^l$ is both of size $F_{out}$ and $F_{in}$.
>
> We clarified this aspect by replacing $F_{in}, F_{out}$ in eq. (1) with $F_{l-1}, F_{l}$, i.e., the embedding sizes at the $l-1$ and $l$-th layer. $F_{l-1}$ can be different from $F_{l}$.

---

> > ### Author Response · Authors · 2024-11-20
> >
> > ### Conclusion
> >
> > We thank the reviewer for pointing out several unclear passages in our work, which we have fixed in the revised version. We hope that this clarifies the doubts and makes the paper easier to follow, as well as the contributions more appreciable. We remain available for further discussion.
> >
> >
> > **References**
> >
> > [1] Saurabh Sihag, Gonzalo Mateos, Corey McMillan, and Alejandro Ribeiro. Covariance neural networks. Advances in neural information processing systems, 2022.
> >
> > [2] Fernando Gama, Elvin Isufi, Geert Leus, and Alejandro Ribeiro. Graphs, convolutions, and neural networks: From graph filters to graph neural networks. IEEE Signal Processing Magazine, 2020.
> >
> > [3] Elvin Isufi, Fernando Gama, David I Shuman, and Santiago Segarra. Graph filters for signal processing and machine learning on graphs. IEEE Transactions on Signal Processing, 2024.
> >
> > [4] Peter J Bickel and Elizaveta Levina. Covariance regularization by thresholding. The Annals of Statistics, 2008.
> >
> > [5] Yash Deshpande and Andrea Montanari. Sparse pca via covariance thresholding. Journal of Machine Learning Research, 2016.
> >
> > [6] Andrea Cavallo, Mohammad Sabbaqi, and Elvin Isufi. Spatiotemporal covariance neural networks. Machine Learning and Knowledge Discovery in Databases. 2024.
> >
> > [7] Alaa Bessadok, Mohamed Ali Mahjoub, and Islem Rekik. Graph neural networks in network neuroscience. IEEE Transactions on Pattern Analysis and Machine Intelligence, 2022.
> >
> > [8] José Vinícius de Miranda Cardoso, Jiaxi Ying, and Daniel Perez Palomar. Algorithms for learning graphs in financial markets, 2020.

---

### Author Response · Authors · 2024-11-20

We wish to thank the reviewers for their effort in reading and analyzing our paper. We are happy to notice that the following strengths of our work have been appreciated:
- The theoretical analysis of the model stability is sound (reviewers rRYr, 9p73)
- The problem of reducing the impact of spurious correlations and improving computational efficiency is significant and our covariance sparsification techniques solve it with theoretical guarantees and experimental evidence (reviewers G6qo, 9p73, rRYr)
- The paper is well-written (reviewers KsmG, rRYr), although some clarifications are needed as we detail in the rebuttals to reviewers 9p73 and G6qo, that we thank for pointing out these unclear passages

Overall, we notice that the weaknesses listed by the reviewers fall under two main categories:
- The first one is related to misunderstandings due to condensed formulations and explanations to fit the page limit. We have addressed these in each specific rebuttal and clarified them in the revised manuscript.
- The second concern, shared by reviewers G6qo, KsmG, rRYr, relates to the novelty of the paper. We would like to present here our motivations why we believe that the contributions of this paper are significant and impactful, not merely incremental.


**Significance of covariance neural networks.**
VNNs [1] are receiving increasing attention [12,13,14,15,16], which makes them an important and relevant topic to study. Furthermore, the VNNs defined in [1] represent a general framework that encompasses several methods commonly applied in practice; indeed, estimating a graph from data correlations and applying a GNN on it is extremely common in fields such as brain connectivity estimation [5,6], financial data processing [7,8] and human action recognition [9,10]. Therefore, the proposed sparsification mechanism of the VNNs can be applied to all these methods for computation reduction and performance improvement in challenging scenarios.


**Significance of sparse settings.**
The problem of estimating a covariance matrix in low-sample regimes, potentially in a sparse setting, is notoriously difficult, and in several cases the basic sample covariance estimator provides very noisy estimates [2,3,4], which hinders the performance of the original VNN in [1]. Since sparsifying the covariance matrix of the data has been shown capable of improving estimation accuracy [2,3,4], and in various applications the data correlations are generally sparse [6,7,8,9,10], we propose sparse covariance neural networks (S-VNNs) that leverage sparsification mechanisms to improve the VNN performance and, in the meantime, enhance the stability and reduce the computation. These findings are corroborated in our experiments in both synthetic and real data.

**Significance of stability analysis.**
Although it is known that the sparsification mechanism has the potential to improve the estimation accuracy of the covariance matrix, its explicit impact on the VNN stability is unknown. Therefore, in addition to proposing S-VNNs, we perform stability analysis to characterize the impact of different sparsification mechanisms on the VNN outputs. The results show that S-VNNs achieve stability to covariance perturbations with a smaller bound compared with the original VNN (see Theorems 2 and 3), and suggest which sparsification to apply based on the problem setting and desired stability. This stability analysis is non-trivial. Indeed, the thresholded covariance estimators in Section 3 do not allow to use convergence results in [17], which are at the basis of the proof in the VNN paper [1], thus calling for a reformulation of the proof. Moreover, the stochastic sparsification in Section 4 requires generalizing the concepts of covariance filter frequency response and integral Lipschitz condition to the case of stochastic filters (see Section 4.1) and extending the stability of GNNs to stochastic perturbations in [11] to the case in which edges are dropped with different probabilities, which is not considered in [11] and significantly changes the problem setting.

**Conclusion**
We believe to have addressed the concerns of the reviewers related to unclear passages in the text, and we hope that our work's contributions are now more clear and evident. We remain available for further questions and discussion.

---

> ### Author Response · Authors · 2024-11-20
>
> **References**
>
> [1] Saurabh Sihag, Gonzalo Mateos, Corey McMillan, and Alejandro Ribeiro. Covariance neural networks. Advances in neural information processing systems, 2022.
>
> [2] Peter J. Bickel and Elizaveta Levina. Regularized estimation of large covariance matrices. The Annals of Statistics 2008.
>
> [3] Peter J Bickel and Elizaveta Levina. Covariance regularization by thresholding. The Annals of Statistics, 2008.
>
> [4] Yash Deshpande and Andrea Montanari. Sparse pca via covariance thresholding. Journal of Machine Learning Research, 2016.
>
> [5] Alaa Bessadok, Mohamed Ali Mahjoub, and Islem Rekik. Graph neural networks in network neuroscience. IEEE Transactions on Pattern Analysis and Machine Intelligence, 2022.
>
> [6] Lishan Qiao, Han Zhang, Minjeong Kim, Shenghua Teng, Limei Zhang, and Dinggang Shen. Estimating functional brain networks by incorporating a modularity prior. NeuroImage, 2016.
>
> [7] José Vinícius de Miranda Cardoso, Jiaxi Ying, and Daniel Perez Palomar. Algorithms for learning graphs in financial markets, 2020.
>
> [8] Yuanrong Wang and Tomaso Aste. Network filtering of spatial-temporal gnn for multivariate timeseries prediction. In Proceedings of the Third ACM International Conference on AI in Finance, 2022
>
> [9] Tianzheng Liao, Jinjin Zhao, Yushi Liu, Kamen Ivanov, Jing Xiong, and Yan Yan. Deep transfer learning with graph neural network for sensor-based human activity recognition. IEEE International Conference on Bioinformatics and Biomedicine, 2022.
>
> [10] Yan Wang, Xin Wang, and Hongmei Yang et al. Mhagnn: A novel framework for wearable sensorbased human activity recognition combining multi-head attention and graph neural networks. IEEE Transactions on Instrumentation and Measurement, 2023.
>
> [11] Zhan Gao, Elvin Isufi, and Alejandro Ribeiro. Stability of graph convolutional neural networks to stochastic perturbations. Signal Processing, 2021.
>
> [12] Saurabh Sihag, Gonzalo Mateos, Corey McMillan, and Alejandro Ribeiro. Explainable brain age prediction using covariance neural networks. Advances in Neural Information Processing Systems, 2024.
>
> [13] Andrea Cavallo, Mohammad Sabbaqi, and Elvin Isufi. Spatiotemporal covariance neural networks. Machine Learning and Knowledge Discovery in Databases. 2024.
>
> [14] Andrea Cavallo, Madeline Navarro, Santiago Segarra, and Elvin Isufi. Fair covariance neural networks, 2024.
>
> [15] Sihag, Saurabh, et al. Transferability of Covariance Neural Networks. IEEE Journal of Selected Topics in Signal Processing, 2024.
>
> [16] Khalafi, Shervin, Saurabh Sihag, and Alejandro Ribeiro. Neural Tangent Kernels Motivate Cross-Covariance Graphs in Neural Networks. Forty-first International Conference on Machine Learning 2024.
>
> [17] Andreas Loukas. How close are the eigenvectors of the sample and actual covariance matrices? International Conference on Machine Learning, 2017.

---

### Meta-Review · Area_Chair_CSiF · 2024-12-18

**Metareview:**

The paper proposes Sparse coVariance Neural Networks (S-VNNs), built upon previous work on VNNs, which applies sparsification techniques on the sample covariance matrix before convolution. When the true covariance matrix is sparse, hard and soft thresholdings are proposed to improve covariance estimation and reduce computational cost. When the true covariance is dense, stochastic sparsification is proposed, where data correlations are dropped in probability according to principled strategies. The proposed method is accompanied by numerical experiments.

While reviewers appreciate that the idea of sparsifying the covariance matrix before convolution is sound, they also have concerns about the limited novelty compared to prior work, the weakness of the theoretical results, and the quality of presentation of the current manuscript (see further below).

**Additional Comments On Reviewer Discussion:**

Among the many concerns raised by the reviewers are the following:

- Revierws G6qo, KsmG, rRYr: the novelty of the paper is limited compared to prior work, especially (Sihag et al 2022). The authors pointed out their contributions using sparse covariance matrices. However, I believe that the current results should be improved (see points below)

- Quality of presentation (Reviewer G6qo): it is difficult to understand the paper without reading the original paper on covariance neural networks (Sihag et al 2022) . The authors have revised the manuscript to address this point but I agree that the exposition in (Sihag et al 2022) is much clearer and the presentation in the current work needs to be considerably improved.

- Reviewer 9p73: In Theorems 2,3, Proposition 1, the probabilities are only stated in the form 1-o(1), instead of being explicitly worked out as in (Sihag et al 2022). The authors pointed out that this is due to the utilization of existing results in the literature. I agree that the current forms of the results are weak.

These points, among many other raised by reviewers, all contribute to the reject decision of the paper.

Further note:

- Reviewer 9p73: Both thresholding and sparsification does not necessarily preserve positive semi-definiteness (PSD) of covariance matrices. The authors pointed out that this is common in the literature. While this is true, it is also a weakness of the proposed approach.
There is a more recent line of work on covariance matrix estimators that preserve positive definiteness, see e.g. Rothman (Biometrika 2012) Positive definite estimators of large covariance matrices.

---

### Decision · Program_Chairs · 2025-01-22

Reject